# Celo: Training Versatile Learned Optimizers on a Compute Diet

**Abhinav Moudgil**[*,1,2]**, Boris Knyazev**[3]**, Guillaume Lajoie**[1,4]**, Eugene Belilovsky**[1,2]
[1]*Mila – Quebec AI Institute,* [2]*Concordia University,* [3]*Samsung – SAIT AI Lab, Montreal,* [4]*Université de Montréal*

**Reviewed on OpenReview:** `https://openreview.net/forum?id=SLqJbt4emY`

## Abstract

Learned optimization has emerged as a promising alternative to hand-crafted optimizers, with the potential to discover stronger learned update rules that enable faster, hyperparameter-free training of neural networks. A critical element for practically useful learned optimizers, that can be used off-the-shelf after meta-training, is strong meta-generalization: the ability to apply the optimizers to new tasks. Recent state-of-the-art work in learned optimizers, VeLO (Metz et al., 2022b), requires a large number of highly diverse meta-training tasks along with massive computational resources, 4000 TPU months, to achieve meta-generalization. This makes further improvements to such learned optimizers impractical. In this work, we identify several key elements in learned optimizer architectures and meta-training procedures that can lead to strong meta-generalization. We also propose evaluation metrics to reliably assess quantitative performance of an optimizer at scale on a set of evaluation tasks. Our proposed approach, Celo[1], makes a significant leap in improving the meta-generalization performance of learned optimizers and also outperforms tuned state-of-the-art optimizers on a diverse set of out-of-distribution tasks, despite being meta-trained for just 24 GPU hours.

## 1  Introduction

Deep learning has advanced remarkably over the past decade by significant improvements in architectures and training techniques, allowing us to train large neural networks on big datasets (Krizhevsky et al., 2012; Achiam et al., 2023; Brooks et al., 2024; Kirillov et al., 2023; Jumper et al., 2021). Much of this progress can be attributed to advancements in *optimizers* (Sutskever et al., 2013; Kingma & Ba, 2015; Loshchilov, 2017; Gupta et al., 2018; Rajbhandari et al., 2020) updating the neural networks (*optimizee*) over the course of training. In deep learning, the most popular optimizers are adaptive variants of stochastic gradient descent such as Adam (Kingma & Ba, 2015; Loshchilov, 2017), AdaGrad (Duchi et al., 2011), Adafactor (Shazeer & Stern, 2018), etc. These variants, although supported by some theoretical guarantees, are often heuristically developed and tested on large-scale neural network tasks (Goodfellow et al., 2016; Schmidt et al., 2021; Bottou et al., 2018; Reddi et al., 2019). In addition to being hand-designed (thus being potentially suboptimal), these optimizers have hyperparameters that need to be tuned for every task, which is quite compute intensive (Schmidt et al., 2021; Sivaprasad et al., 2020). Additionally, recent works also use *schedules* (Goyal, 2017; You et al., 2017; Smith, 2017; Loshchilov & Hutter, 2016) which adjust optimizer hyperparameters dynamically throughout training. These schedules introduce additional tunable hyperparameters, making the whole training process even more computationally expensive (Metz et al., 2020b; Dahl et al., 2023).

Deep learning advances allowed us to reduce feature engineering efforts and to greatly improve performance by replacing hand-crafted features (Lowe, 2004) with learned ones. Inspired by the advent of deep learning, learned optimizers present a promising avenue to make the optimization process hyperparameter free

---

*Correspondence to: Abhinav Moudgil <`abhinav.moudgil@mila.quebec`>
[1]Code is available at: `https://github.com/amoudgl/celo`

and also more performant (Andrychowicz et al., 2016; Wichrowska et al., 2017; Almeida et al., 2021; Metz et al., 2022a;b). Formally, given an optimizee neural network with $n$ parameters $\boldsymbol{\theta}_t \in \mathbb{R}^n$ at iteration $t$, an optimizer $f_\phi$, parameterized by $\phi$ (e.g. hyperparameters), takes as input the gradient $\nabla \boldsymbol{\theta}_t$, current state $\boldsymbol{s}_t$ (e.g. momentum) and returns updated parameters $\boldsymbol{\theta}_{t+1}$ along with updated state $\boldsymbol{s}_{t+1}$: $(\boldsymbol{\theta}_{t+1}, \boldsymbol{s}_{t+1}) = f_\phi(\boldsymbol{\theta}_t, \nabla \boldsymbol{\theta}_t, \boldsymbol{s}_t)$. While in a hand-designed optimizer an update rule and $\phi$ are predefined by the practitioner, a learned optimizer $f_\phi$ *learns* the entire update rule parameterized by $\phi$ through a meta-training process. Learning the update rule allows learned optimizers to leverage additional meta-data $\mathcal{M}_t$, such as training loss (Metz et al., 2020a; 2022b), training progress (Almeida et al., 2021; Metz et al., 2022a;b), neural network graph (Peebles et al., 2022; Kofinas et al., 2024)) and other relevant information. Thus, $f_\phi$ can learn a more powerful update rule formalized as the following:

$$(\boldsymbol{\theta}_{t+1}, \boldsymbol{s}_{t+1}) = f_\phi(\underbrace{\boldsymbol{\theta}_t, \nabla \boldsymbol{\theta}_t, \mathcal{M}_t}_{\boldsymbol{\psi}_t}, \boldsymbol{s}_t), \tag{1}$$

where $\boldsymbol{\psi}_t$ represents all the "features" of the optimizee network at iteration $t$. Hence, learned optimizers can automatically change the update function or implement schedules by monitoring the input meta-data over the course of training, thus removing the expensive manual tuning of optimizer hyperparameters.

Despite the potential of learned optimizers, one of their major challenges is *meta-generalization*: learned optimizers once meta-trained on a distribution of optimization tasks should be able to generalize to unseen or out-of-distribution tasks. Prior work (Metz et al., 2020a; 2022b) has shown that akin to scaling laws in deep learning (Kaplan et al., 2020; Achiam et al., 2023), the meta-generalization of these learned optimizers improves as meta-training is scaled up. However, the meta-training of these learned optimizers as in prior work, VeLO (Metz et al., 2022b), is prohibitively expensive (4000 TPU months) which poses a serious challenge in further improvements of these learned optimizers and highlights the need for more compute-efficient approaches (Rezk et al., 2023; Metz et al., 2022b). Therefore, in this work, we keep a small fixed meta-training set and instead identify several elements in the design and meta-training of these learned optimizers which could lead to strong meta-generalization while using only a fraction of VeLO's compute budget (24 GPU hours).

Another challenge in learned optimization is *quantitative* evaluation, which can be used as a reliable measure to track progress in this field. Most prior work in learned optimizers (Schneider et al., 2019; Schmidt et al., 2021; Metz et al., 2022b;a) focus on loss curves or final performance of the trained models on selected tasks as the primary mode of evaluation. However, it remains unclear how to obtain a reliable *aggregated* performance of an optimizer across a diverse, large set of tasks, each with potentially different evaluation metrics and loss scales. Furthermore, learned optimizers often also exhibit high variance and occasional instability across tasks (Andrychowicz et al., 2016; Metz et al., 2022a;b) which can also significantly bias the evaluation results due to outliers. In this work, we seek inspiration from Reinforcement Learning (Agarwal et al., 2021), which also faces similar evaluation challenges and proposes reliable evaluation metrics which we employ to quantitatively evaluate learned optimizers. Our contributions in this work are the following:

1. We propose a simple recipe to train meta-generalizable (versatile) learned optimizers on a limited compute budget; our recipe's main ingredients are task augmentation, simple design consisting of per-param learned update rule with a high-level scheduler and decoupled training of the update rule and scheduler.

2. We provide reliable evaluation metrics to track aggregate performance of optimizers on a set of evaluation tasks which are robust to outliers.

3. Our proposed approach, Celo, outperforms 15 hand-designed optimizers, including Adam (Kingma & Ba, 2015) and Shampoo (Anil et al., 2020), as well as state-of-the-art learned optimizers such as VeLO (Metz et al., 2022b) on a diverse set of out-of-distribution tasks.

4. We conduct an exhaustive ablation study over each component in our approach with meta-generalization as the principal focus.

5. We analyze schedules predicted by Celo and find that they are adaptive to unseen tasks and training horizons, further highlighting the meta-generalization capabilities of Celo.

## 2 Background

**Learned MLP update rule.** Several learned optimizer architectures have been proposed in previous works (Andrychowicz et al., 2016; Metz et al., 2020a; Almeida et al., 2021; Metz et al., 2022a;b), each varying in the types of inputs they digest, the elements they maintain within their state, and the functional form of their outputs. In this section, we briefly review a learned optimizer parameterized using a multi-layer perceptron (MLP) (Metz et al., 2022a) (Adafac MLP LOpt) that our work and recent state-of-the-art works like VeLO (Metz et al., 2022b) also build upon. It is a simple MLP-based learned optimizer that can serve as a drop-in replacement for hand-designed update rule such as SGD or Adam. Concretely, at a given iteration $t$, Adafac MLP LOpt takes as input parameter vector $\boldsymbol{\theta}_t$, gradient $\nabla \boldsymbol{\theta}_t$ along with current iteration $t$. These values are then used to update momentum accumulator buffers in state $\boldsymbol{s}_t$. These accumulated values are then used to build input features $\boldsymbol{F}$ for each parameter which are passed to the learned MLP rule along with the global training progress features $\boldsymbol{\omega}^p$ (see (Metz et al., 2022a) for all the input features). The learned MLP returns two outputs of the same dimensionality as $\boldsymbol{\theta}_t$, corresponding to direction $\boldsymbol{d}$ and magnitude $\boldsymbol{m}$, which are used to get parameter updates $\Delta \boldsymbol{\theta}_t$ and then parameters $\boldsymbol{\theta}_{t+1}$ as follows:

$$
\begin{aligned}
\Delta \boldsymbol{\theta}_t &= \lambda_1 \boldsymbol{d} \cdot e^{(\lambda_2 \boldsymbol{m})} \\
\boldsymbol{\theta}_{t+1} &= \boldsymbol{\theta}_t - \Delta \boldsymbol{\theta}_t,
\end{aligned} \tag{2}
$$

where $\lambda_1$ and $\lambda_2$ are fixed scalars set to low values (0.001) in order to keep meta-training stable. VeLO maintains multiple such learned MLPs (of the same architecture) in its bank which are combined to give a weight update at each step. In addition to learned MLPs that are applied to each parameter (per-parameter MLP), VeLO has a per-tensor LSTM network applied to each tensor (e.g. layer weight or bias). The per-tensor LSTM takes global features such as training progress, loss and per-tensor features such as variance of momentum, mean of gradient RMS, etc. Furthermore, VeLO scales per-parameter updates with a tensor norm of the corresponding layer. VeLO's more sophisticated hierarchical design makes it more performant than Adafac MLP LOpt while keeping it efficient at run-time due to its hierarchical design. Specifically, the hierarchical design allows adding capacity at higher levels, such as the layer (or tensor) and global levels, whose added computational costs scale *sublinearly* with the number of parameters. In this work, we also propose a hierarchical architecture similar to VeLO but is much simpler in design and meta-generalizes significantly better than VeLO and Adafac MLP LOpt.

**Meta-training.** Unlike hand-designed optimizers, learned optimizers' update function is parametrized by $\boldsymbol{\phi}$ learned through a meta-training process. A standard approach to meta-training involves solving a bi-level optimization problem that involves an inner problem which optimizes network parameters $\boldsymbol{\theta}$ using the learned optimizer update $\boldsymbol{\phi}$ on a sampled task and an outer problem which optimizes the learned optimizer parameters $\boldsymbol{\phi}$ based on the feedback from the inner loop (Andrychowicz et al., 2016; Wichrowska et al., 2017; Metz et al., 2019a; 2020a; Vicol et al., 2021). Formally, given a set of optimization tasks $\mathcal{T}$, the learned optimizer parameters $\boldsymbol{\phi}$ are obtained by sampling an optimization task which consists of data distribution $\mathcal{D}$, initial network parameters $\boldsymbol{\theta}_0$, and a training objective $\mathcal{L}$ and solving the bi-level problem below:

$$
\boldsymbol{\phi}^* = \arg \min_{\boldsymbol{\phi}} \mathbb{E}_{(\mathcal{D}, \mathcal{L}, \boldsymbol{\theta}_0) \sim \mathcal{T}} \mathbb{E}_{X_t \sim \mathcal{D}} \left( \frac{1}{T} \sum_{t=0}^{T-1} \mathcal{L}(X_t; \boldsymbol{\theta}_t, \boldsymbol{\phi}) \right), \tag{3}
$$

where the inner loop is recursively defined for $t = [0, T-1]$ as:

$$
\begin{aligned}
(\boldsymbol{\theta}_t, \boldsymbol{s}_t) &= (\boldsymbol{\theta}_0, \mathbf{0}) && \text{if } t = 0, && (4) \\
(\boldsymbol{\theta}_t, \boldsymbol{s}_t) &= f_{\boldsymbol{\phi}}(\boldsymbol{\theta}_{t-1}, \nabla \boldsymbol{\theta}_{t-1}, \mathcal{M}_{t-1}, \boldsymbol{s}_{t-1}) && \text{if } t > 0; && (5) \\
\nabla \boldsymbol{\theta}_t &= \frac{\partial \mathcal{L}(X_t; \boldsymbol{\theta}_t, \boldsymbol{\phi})}{\partial \boldsymbol{\theta}_t}; \quad X_t \sim \mathcal{D} && \forall t. && (6)
\end{aligned}
$$

Here $T$ denotes the unroll length in the inner loop and $X$ denotes sampled data from $\mathcal{D}$. The outer training objective or meta-objective is based on the mean loss as formalized above or, less common, the final loss of the inner loop (Metz et al., 2019a; 2020a; 2022b). We compute meta-gradients for $\boldsymbol{\phi}$ in eq. 3 using Persistent

Evolution Strategies (PES) (Vicol et al., 2021) used in prior work (Metz et al., 2022a; Harrison et al., 2022; Gärtner et al., 2023) since directly back-propagating through the inner loop in eq. 3 can lead to noisy gradient estimates, especially when the unroll length $T$ is large (Metz et al., 2019a; Vicol et al., 2021). Moreover, PES also gives unbiased gradient estimates in the truncated unroll setting in which the learned optimizer is updated in truncations every few steps in the inner loop instead of updating only once after the full unroll of the inner loop. This allows frequent updates to the optimizer during meta-training, thus making it less computationally expensive than full unrolls (Vicol et al., 2021; Metz et al., 2022b).

## 3 Celo: Compute-efficient learned optimizer

---
**Algorithm 1** Celo update

---

**Input:**    $\boldsymbol{\theta}_t$     optimizee parameters
           $\nabla\boldsymbol{\theta}_t$    gradients
           $L_t$      current loss
           $\boldsymbol{s}_t$      optimizer state
**Require:**    learned scheduler $f_{\text{lstm}}$, learned update rule $f_{\text{mlp}}$,
              accumulators update function $f_{\text{acc}}$, fixed scalars $\alpha, \lambda_1, \lambda_2 \in \mathbb{R}$

  1:   $\boldsymbol{s}_{t+1} \leftarrow f_{\text{acc}}(\boldsymbol{s}_t, \nabla\boldsymbol{\theta}_t, L_t, t)$                           $\triangleright$ update accumulators in state
  2:   compute progress features $\boldsymbol{\omega}^p$, loss features $\boldsymbol{\omega}^l$ using updated $\boldsymbol{s}_{t+1}$
  3:   $\boldsymbol{x}^s \leftarrow \text{concat}(\boldsymbol{\omega}^p, \boldsymbol{\omega}^l)$                                        $\triangleright$ scheduler input
  4:   $o_t \leftarrow f_{\text{lstm}}(\boldsymbol{x}^s)$                                     $\triangleright$ learned scheduler forward pass
  5:   $\eta_t \leftarrow \alpha e^{o_t}$                                          $\triangleright$ scheduler output
  6:   initialize next params $\boldsymbol{\theta}_{t+1} \leftarrow ()$
  7:   **for** each tensor with params $\boldsymbol{p}_t \in \boldsymbol{\theta}_t$ in parallel **do**       $\triangleright$ parallel scan over all layer weights & biases
  8:      prepare per-param features $\boldsymbol{F}$ for $\boldsymbol{p}_t$ using updated state $\boldsymbol{s}_{t+1}$
  9:      $\boldsymbol{d}, \boldsymbol{m} \leftarrow f_{\text{mlp}}(\boldsymbol{F})$                            $\triangleright$ learned update rule forward pass
10:      $\Delta\boldsymbol{p}_t \leftarrow \lambda_1\boldsymbol{d} \cdot e^{(\lambda_2\boldsymbol{m})}\|\boldsymbol{p}_t\|_2$                 $\triangleright$ compute per-param updates
11:      $\Delta\boldsymbol{p}_t \leftarrow \eta_t\Delta\boldsymbol{p}_t$                              $\triangleright$ scale by scheduler output
12:      $\boldsymbol{p}_{t+1} \leftarrow \boldsymbol{p}_t - \Delta\boldsymbol{p}_t$                           $\triangleright$ updated params
13:      $\boldsymbol{\theta}_{t+1} \leftarrow \boldsymbol{\theta}_{t+1} \cup \boldsymbol{p}_{t+1}$                       $\triangleright$ gather params
     **return** $\boldsymbol{\theta}_{t+1}, \boldsymbol{s}_{t+1}$

---

In this work, we treat meta-generalization as the principal component for designing and training learned optimizers. We show that our proposed approach, *Celo* (short for "**C**ompute-**e**fficient **l**earned **o**ptimizer"), incorporating a recipe of key algorithmic meta-training and architectural improvements yields strong generalization to unseen tasks. Our meta-training recipe described below can be followed on small-scale compute budget (more details in §5) and experiments in §6 show that each element is critical for strong meta-generalization:

**Task augmentation.**   Building on the success of data augmentation strategies in deep learning (Krizhevsky et al., 2012; Cubuk et al., 2020; Chen et al., 2020b; Laskin et al., 2020), we apply task augmentation during meta-training to enhance the robustness of learned optimizers (Metz et al., 2022b). This approach emulates learning dynamics across a potentially large number of tasks using a limited set of meta-training tasks (Metz et al., 2022b). Task augmentation is done by re-scaling parameters (re-parametrizing) in the inner loop. Recall that each meta-training iteration consists of applying an inner training loop on a randomly initialized, optimizee network with parameters, $\boldsymbol{\theta}_0$. A random scalar parameter $\tau$ is sampled in each meta-training iteration by which the optimizee network parameters at each inner training step $t$ are scaled by $\tau$ before the forward pass as $\tau\boldsymbol{\theta}_t$ (re-parametrization) which yields inner task loss $\mathcal{L}$. At initialization, the initial weights $\boldsymbol{\theta}_0$ are also scaled by a factor of $1/\tau$ in order to keep the underlying optimizee function output same. Modified Eq. 3 after task augmentation is given below with task augmentation specific changes highlighted in red:

$$\arg\min_{\boldsymbol{\phi}} \mathbb{E}_{(\tau, \mathcal{D}, \mathcal{L}, \boldsymbol{\theta}_0) \sim \mathcal{T}} \mathbb{E}_{X_t \sim \mathcal{D}} \left( \frac{1}{T} \sum_{t=0}^{T-1} \mathcal{L}\big(X_t; \tau\boldsymbol{\theta}_t, \boldsymbol{\phi}\big) \right), \tag{7}$$

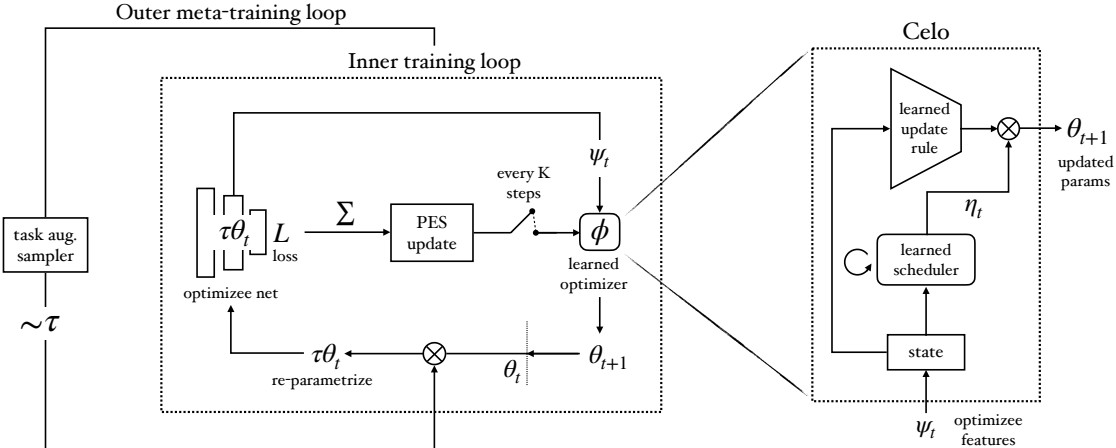

Figure 1: **Celo meta-training and architecture.**

where in the inner loop for $t = [0, T-1]$:

$$(\boldsymbol{\theta}_t, \boldsymbol{s}_t) = (\boldsymbol{\theta}_0/\tau, \mathbf{0}) \qquad \qquad \text{if } t = 0, \qquad (8)$$

$$(\boldsymbol{\theta}_t, \boldsymbol{s}_t) = f_\phi(\boldsymbol{\theta}_{t-1}, \nabla\boldsymbol{\theta}_{t-1}, \mathcal{M}_{t-1}, \boldsymbol{s}_{t-1}) \qquad \qquad \text{if } t > 0; \qquad (9)$$

$$\nabla\boldsymbol{\theta}_t = \frac{\partial \mathcal{L}(X_t; \tau\boldsymbol{\theta}_t, \phi)}{\partial \boldsymbol{\theta}_t}; \quad X_t \sim \mathcal{D} \qquad \qquad \forall t. \qquad (10)$$

Overall, this task augmentation does not change the underlying optimizee network output at initialization, but it does affect the learning dynamics of the optimizee network in the inner loop when it is updated with adaptive or learned optimizers, thus effectively "simulating" more tasks from a fixed amount of given meta-training tasks (see Fig. 2 for a representative example).

**Simple learned optimizer design: learned scheduler with a learned update rule.** Currently, the most performant and dominant paradigm in the optimization of modern deep neural networks is to use a *scheduler* coupled with a hand-designed optimizer (Goyal, 2017; You et al., 2017; Smith, 2017; Loshchilov & Hutter, 2016). A scheduler changes globally the learning rate of the underlying hand-designed optimizer as training progresses. Learning rate schedules have a long history in non-adaptive stochastic gradient descent methods to address the high variance of the gradient estimator close to the optimum (Boyd & Vandenberghe, 2004). Deep Learning practitioners have shown that sophisticated schedules such as warm-ups and cyclic learning rate (Smith, 2017; Loshchilov & Hutter, 2016; Touvron et al., 2023) help significantly in the optimization of deep neural networks and

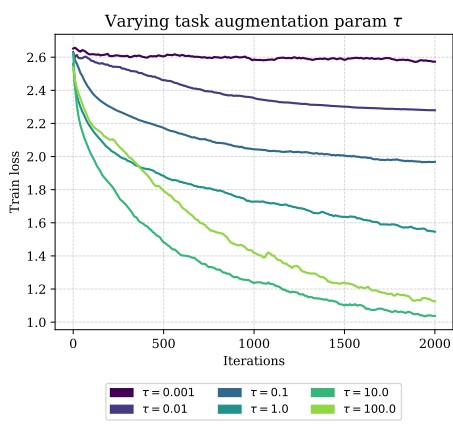

Figure 2: **Task augmentation.** Learning curves for Adam with fixed (1e-4) learning rate with different augmentation parameters ($\tau$) for a CIFAR-10 ConvNet task.

empirically, it has been shown that schedulers are beneficial even in the case of adaptive optimizers such as Adam (Loshchilov, 2017; Touvron et al., 2023). Inspired by this, we focus on separating the global learning rate scheduler from the per-parameter update rule.

Several designs for learned optimizers have been proposed in prior work which extend this notion of schedulers beyond a scalar learning rate (Andrychowicz et al., 2016; Wichrowska et al., 2017; Almeida et al., 2021; Peebles et al., 2022; Metz et al., 2022b), see Fig. 8 in Appendix. However, in this work, we streamline the learned optimizer design and show that a learned recurrent scheduler (LSTM in our case) which outputs a

scalar schedule parameter coupled with learned per-parameter MLP update rule generalizes well to a large-number of unseen tasks, when even meta-trained on a limited number of small-scale tasks. Our learned LSTM-based scheduler takes global loss and progress features (Metz et al., 2022b;a) as input and outputs a positive scalar $\eta_t$ which is obtained by linearly projecting the LSTM output and raising it to the exponential. We show that this exponential form in scheduler is crucial for performance in §5. The final parameter updates are obtained by scaling the per-parameter MLP (Metz et al., 2022a;b) outputs with this scalar $\eta_t$ predicted by the scheduler. The pseudocode for forward pass of our optimizer is expressed concretely in Algorithm 1.

**Decoupled training of scheduler and update rule.** We propose to view an optimization problem in two parts: how to control step-size and what rule to use for parameter updates. Current state-of-the-art hand-designed optimizers such as Adam (Kingma & Ba, 2015) and Shampoo (Gupta et al., 2018) propose a hand-designed update rule, leaving the task of controlling or modulating global learning rate and other hyper-parameters up to the practitioners. Following this view, we propose to meta-train learned optimizers in a two-stage process: (1) Learn a strong parameter update rule (2) Freeze the learned update rule and learn a step-size scheduler. Our experiments show that two-stage training is crucial for meta-generalization and avoids meta-overfitting when training on a compute budget with fixed amount of meta-training tasks, hence outperforming single-stage training. Moreover, our simple two-stage design allows us to contrast and compare with state-of-the-art hand-designed optimiz-

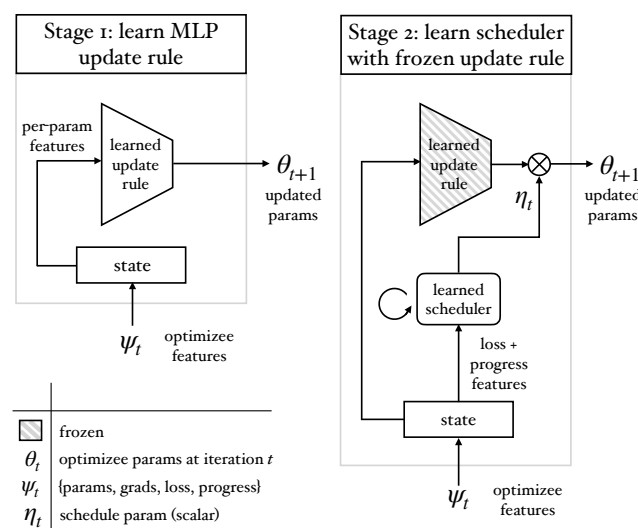

Figure 3: **Two-stage training of Celo.**

ers. Our experiments show that our learned update rule outperforms hand-designed update rules like Adam and when combined with a learned scheduler, the meta-generalization performance further improves.

## 4 Reliably scoring optimizers at scale

Prior work in learned optimizers often resorts to showing training or validation curves as the primary means of evaluation (Schneider et al., 2019; Schmidt et al., 2021; Metz et al., 2022b;a). This is a common practice, because evaluating the performance of learned optimizers quantitatively is challenging. For example, different evaluation tasks can have different ranges and semantics of loss values. Moreover, learned optimizers also often show unstable behaviour and evaluating them multiple times to reduce the variance can be infeasible, since each evaluation run involves training a neural network. Therefore, metrics such as mean or median (of loss) can be noisy. Alternatively, an aggregate speedup can be reported, as done in prior work (Dahl et al., 2023). However, this approach requires carefully designing target scores for each task and meaningfully aggregating speedups, which becomes cumbersome for a large and diverse set of potentially changing tasks. Moreover, if the targets are set too high, the optimizers that fail to meet these targets receive a zero score, disregarding improvements between the optimizers. Conversely, setting targets too low introduces a large bias from outliers, compromising the evaluation's reliability.

Hence, in this work, we address the following problem: *How do we reliably compute aggregated score for each optimizer at scale without hand-designing any thresholds?* We find that this evaluation problem is analogous to the one in Reinforcement Learning (RL) in which an RL agent is evaluated on several tasks, e.g. Atari games. Agarwal et al. (2021) propose a widely adopted solution to this evaluation problem (Schwarzer et al., 2023; Ceron et al.; Micheli et al., 2022). Specifically, they show that the inter-quartile mean (IQM) accurately aggregates results with only a few evaluation runs and is less sensitive to outliers than other metrics. For

evaluating optimizers, we focus on two aspects: final loss performance and speedup with respect to a tuned baseline.

**Final loss IQM.** Given a fixed set of $M$ tasks and $N$ trials for each task, IQM (Agarwal et al., 2021) expects $M \times N$ normalized score matrix $\boldsymbol{S}$ for each algorithm. In the RL setup, Agarwal et al. (2021) compute a scalar normalized score $s_{m,n} \in \boldsymbol{S}$ for the $m$-th game and $n$-th trial by scaling the game score in that trial with respect to the human score on the corresponding game. This implies that, in any trial, a normalized score $s_{m,n} > 1$ represents "Super-Human" performance. Equivalently, in optimization, we treat the best Adam run from a fixed number of trials as the "human baseline". The normalized score for each run $s_{m,n}$ is computed as the ratio between the final loss of the best Adam run $L_T^{\mathrm{adam}}$ after optimizing for $T$ steps, and the final loss obtained by the optimizer being evaluated ($L_T^{\mathrm{opt}}$):

$$s_{m,n} = \frac{L_T^{\mathrm{adam}}}{L_T^{\mathrm{opt}}}. \tag{11}$$

where $s_{m,n} > 1$ represents "Super-Adam" performance. The $M \times N$ normalized scores are then used to compute the aggregate IQM metric. Specifically, IQM is computed by discarding top & bottom 25% scores and calculating the (trimmed) mean of the remaining 50% of the normalized scores.

**Speedup IQM.** Similar to the setup described above, we first compute a normalized speedup $M \times N$ normalized score matrix $\boldsymbol{S}$ which is then used to compute the aggregate speedup IQM metric. We compute normalized scores as follows: For a given task, we first pick the best Adam baseline out of $K$ trials which achieves the lowest loss at the end of training, which consists of say, $T$ steps. Next, we check the number of steps for the optimizer being evaluated to achieve the same loss, say, $T^{\mathrm{opt}}$ and compute normalized speedup score $s_{m,n}$ as below:

$$s_{m,n} = \frac{T}{T^{\mathrm{opt}}}. \tag{12}$$

If an optimizer is unable to reach the final loss achieved by the best tuned Adam baseline in any run, we set its corresponding speedup score to zero assuming $T^{\mathrm{opt}} = \infty$.

## 5 Experiments

### 5.1 Baselines

We compare our proposed optimizer, Celo, with 15 state-of-the-art hand-crafted optimizers and 4 learned optimizers. Among the learned optimizers, we compare with: (1) NNAdam LOpt (Metz et al., 2020b), a hybrid-optimizer using Adam as the update rule whose hyperparameters are controlled by a learned LSTM-based module; (2) AdaFac MLP LOpt (Metz et al., 2022a) described in Section 2; (3) RNN MLP LOpt (Metz et al., 2020a) using an LSTM-based per-tensor network that communicates with a learned MLP update rule through embeddings; (4) VeLO (Metz et al., 2022b) with an LSTM-based per-tensor network that generates weights of an MLP update rule through a hyper-network. Celo and other RNN-based optimizers use a hidden size of 64 in their recurrent networks. VeLO's default hidden size is 512, therefore to focus exclusively on differences in architecture, as a baseline we use "VeLO-S" with hidden size reduced to 64 and accordingly the number of MLPs in its bank reduced from 256 to 32. This scaling down does not severely impact meta-generalization (see Appendix A.2). Among the hand-crafted optimizers, we compare with Adam (Kingma & Ba, 2015), Shampoo (Anil et al., 2020; Gupta et al., 2018) with SGD and Adagrad grafting (default block size 128), Nesterov accelerated AdamW (NAdamW) (Dozat, 2016; Loshchilov, 2017), RAdam (Liu et al., 2019), Yogi (Zaheer et al., 2018), AdaBelief (Zhuang et al., 2020), LAMB (You et al., 2019), LARS (You et al., 2017), RMSProp (Tieleman & Hinton, 2012), Adafactor (Shazeer & Stern, 2018), AdaGrad (Duchi et al., 2011), SM3 (Anil et al., 2019), SGD with momentum (Sutskever et al., 2013) and Fromage (Bernstein et al., 2020) for which there is released baseline data from VeLO with 15 trials per task tuning learning rate logarithmically with half powers of 10. Morever, NAdamW, which serves as a superset of many adaptive optimizers (Choi et al., 2019; Dahl et al., 2023), is aggressively tuned with 1000 trials searching over different configurations of learning rate, $\beta_1$, $\beta_2$, $\epsilon$ and cosine learning rate schedule (Metz et al., 2020b). We pick the best trial based on final loss for evaluation and visualization.

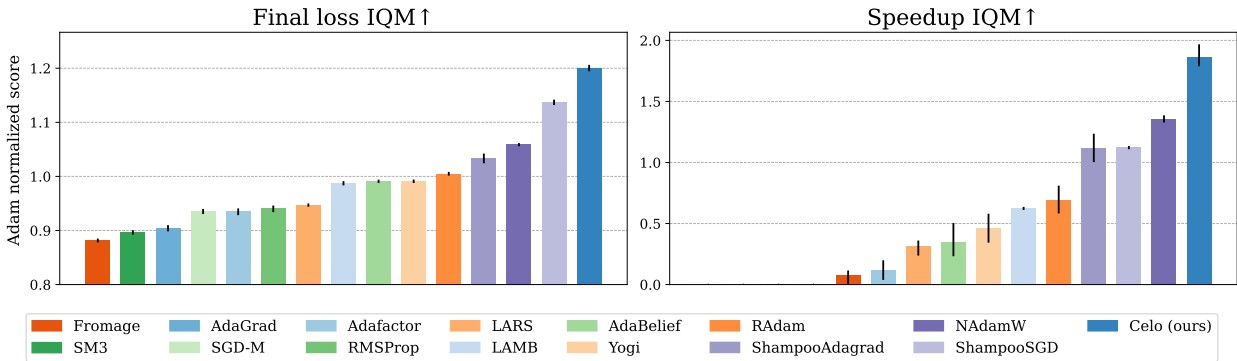

Figure 4: **Comparing Celo with state-of-the-art hand-crafted optimizers.** Celo is our proposed learned optimizer meta-trained on a limited budget of 24 GPU hours. Optimizers are evaluated with final loss (left) and speedup (right) criteria with respect to Adam on a diverse set of 17 tasks which are out-of-distribution for Celo and include image classification, language modeling, autoencoders, learned optimizer training, etc. IQM score above 1.0 indicates "Super-Adam" performance, read more about our evaluation methodology in §4 and meta-training/evaluation tasks in §5.

## 5.2 Meta-training

All the learned optimizers are meta-trained with the same setup on a *fixed* compute budget i.e. given a fixed set of meta-training tasks, we study how far can we push the meta-generalization performance of learned optimizers. We take a set of four meta-training tasks proposed by Metz et al. (2022b) for all our experiments which contains four image-classification datasets including MNIST (LeCun & Cortes, 1998), Fashion-MNIST (Xiao et al., 2017), SVHN (Netzer et al., 2011), and CIFAR-10 (Krizhevsky et al., 2009) paired with a one-layer MLP network with 32 hidden units and ReLU activations. Additionally, images are resized to 8×8 to keep the meta-training computationally efficient and fast like in Metz et al. (2022b). All the learned optimizers are meta-trained with truncated PES (Vicol et al., 2021) with maximum 2K inner unroll length for 100K meta-iterations using mean loss of the inner loop as the meta-objective following prior work (Metz et al., 2022a; Vicol et al., 2021; Harrison et al., 2022). With this setup, all our meta-training experiments finish in a day (<24 hours) on a single Nvidia RTX8000 GPU. Note that we intentionally did not scale up meta-training here in order to do an exhaustive controlled study within our compute budget and solely focus on improving the meta-generalization performance. For task augmentation (§3), we sample $\tau$ uniformly on log scale between 0.001 and 1000 in each meta-iteration. See further implementation details in Appendix A.3.

## 5.3 Evaluating on a broad spectrum of tasks

We test all the optimizers on 17 diverse unseen tasks with 3 seeds per task. These tasks span different machine learning problems (including classification, language modeling, learned optimizer training, auto-encoders) with different architecture types like MLPs, ConvNets, Transformers, RNNs, etc., and varying model sizes. We pick this evaluation set from a larger VeLOdrome task set (Metz et al., 2022b) of 83 tasks in order to fit evaluation of all the experiments within our compute budget. The tasks are described below:

**Image MLPs.** Total 3 image MLP tasks. Two CIFAR-10 tasks: 3-layer MLP with 128 hidden units, layer-norm and ReLU activations after each layer; 3-layer MLP with 128 hidden units and Tanh activations. One FashionMNIST task with a 2-layer MLP of size 128 and ReLU activations.

**CNNs.** Total 4 convolutional neural network (CNN) tasks. One CIFAR-100 task with three convolution layers having 32, 64 and 64 channels respectively. Three CIFAR-10 tasks with the same (32-64-64) layers as in CIFAR-100, but different normalization schemes: (1) layer-norm (2) batch-norm (3) without any normalization. All these tasks use ReLU activations.

| w/ task augmentation | final loss | | | speedup | | |
|---|---|---|---|---|---|---|
| | median↑ | OG↓ | IQM↑ | median↑ | OG↓ | IQM↑ |
| nnadam lopt (Metz et al., 2020b) | 0.69 | 0.39 | 0.67 | 0.00 | 0.82 | 0.00 |
| rnn mlp lopt (Metz et al., 2020a) | 1.00 | 0.08 | 1.00 | 1.07 | 0.45 | 0.69 |
| adafac mlp lopt (Metz et al., 2022a) | 1.04 | 0.02 | 1.05 | 1.32 | 0.24 | 1.30 |
| velo-s (Metz et al., 2022b) | 0.93 | 0.17 | 1.00 | 0.00 | 0.53 | 0.81 |
| celo (ours) | **1.14** | **0.01** | **1.20** | **1.92** | **0.14** | **1.86** |

Table 1: **Comparing Celo with learned optimizers.** All the learned optimizers are meta-trained on the same set of 4 tasks with task augmentation and evaluated on 17 diverse out-of-distribution tasks (listed in §5). IQM score above 1.0 indicates "Super-Adam" performance (read more in §4). Our learned optimizer, Celo, incorporating the recipe proposed in §3 significantly improves meta-generalization.

**ViT.** Two vision transformer (ViT) (Dosovitskiy et al., 2021) tasks on CIFAR-100: (1) "wide shallow" with 6 layers, 6 heads and hidden size 384; (2) "skinny deep" with 10 layers, 4 heads and hidden size 128. Both ViT variants represent images as $16 \times 16$ patches.

**Transformer LM.** Three transformer decoder language modeling tasks (Radford et al., 2019) on LM1B (Chelba et al., 2013) of an increasing scale: (1) hidden size 20, 5 heads, 1 layer, sequence length 8; (2) hidden size 32, 4 heads, 2 layer, sequence length 8; (3) hidden size 32, 4 heads, 2 layer, sequence length 32.

**RNN LM.** Two recurrent neural network (RNN) language modeling tasks on the LM1B32k (Brants et al., 2007; Chelba et al., 2013) and wikipedia32k (Merity et al., 2016) datasets. Both tasks have the same RNN: an LSTM with hidden size 256, input embedding size 128 and sequence length 32.

**Auto-encoders.** Two image MLP autoencoder tasks: (1) 3-layer 128-32-128 MLP on CIFAR10 and batch size 256 (2) 3-layer 128-32-128 MLP on MNIST and batch size 128.

**Learning Optimizers.** One learned optimization task to test the ability of our proposed optimizer to train new learned optimizers. This task meta-trains a learned optimizer with an MLP-based optimizer architecture on FashionMNIST with the maximum inner unroll length 50 and PES as the meta-gradient estimator. The optimizee network is a 1-layer MLP with 32 hidden units.

### 5.4 Evaluation metrics

All the 17 tasks from our suite are first optimized using Celo and the baselines for 2K iterations. We then evaluate them using the proposed final loss and speedup IQM (Section 4). Additionally, we report aggregate Median and Optimality Gap (OG) metrics from Agarwal et al. (2021) for both criteria, final loss and speedup, using normalized score matrix $S$ defined in §4. The optimality gap metric (Agarwal et al., 2021) is computed by first clipping all the normalized scores above 1.0 (optimal baseline score) to 1.0 and then subtracting the mean of all the clipped scores from the optimal baseline performance score 1.0.

## 6 Results

**Celo generalizes to unseen tasks.** Figure 4 and Table 1 show the evaluation performance of our Celo on the 17 unseen tasks along two criteria: final loss and speedup. For both criteria, Celo outperforms all the previous learned optimizer approaches as well as standard hand-crafted optimizers such as Shampoo and NAdamW. This result is remarkable for two reasons. First, these evaluation tasks are much larger in scale in terms of flops and parameter count and far out-of-distribution w.r.t what Celo has seen during meta-training. Second, all the hand-crafted optimizers use 10-1000s of tuning trials per task, out of which the best trial is picked for evaluation. Notably, for speedup, Celo achieves IQM 1.86 and outperforms all the baselines by a significant margin. Moreover, even after applying task augmentation in all the learned optimizer baselines, making them stronger (Table 2a), Celo leads by a large margin (Table 1) which further highlights the generalization strength of Celo to out-of-distribution tasks from limited meta-training.

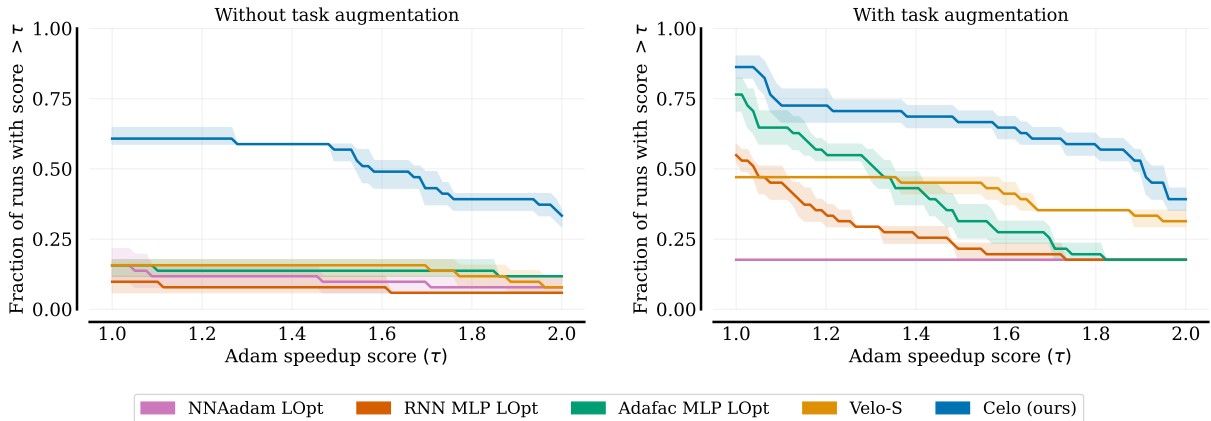

Figure 5: **Meta-generalization speedup profiles.** Task augmentation improves meta-generalization performance in terms of speedup for all the learned optimizers. Celo outperforms all the learned optimizer baselines with and without task augmentation.

| optimizer | IQM↑ | |
|---|---|---|
| | w/o aug | w/ aug |
| nnadam lopt | 0.65 | **0.67** |
| rnn mlp lopt | 0.73 | **1.00** |
| adafac mlp lopt | 0.89 | **1.05** |
| velo-s | 0.90 | **1.01** |
| celo | 1.07 | **1.20** |

(a)

| | task aug | | | |
|---|---|---|---|---|
| | update rule | scheduler | IQM↑ | OG↓ |
| celo w/o aug | | | 1.07 | 0.04 |
| | ✓ | | 1.16 | 0.04 |
| | | ✓ | 1.10 | 0.08 |
| celo | ✓ | ✓ | **1.20** | **0.01** |

(b)

Table 2: **Task augmentation.** Task augmentation improves generalization across the board for all learned optimizer architectures (a) and helps the most when it is used at both stages of Celo meta-training (b).

**Task augmentation is key for generalization.** Table 2a and Figure 5 show the impact of using task augmentation for all the learned optimizer baselines and Celo. It is important to note that we use the same meta-training dataset and algorithm for all the learned optimizers and only ablate over task augmentation. As evident from the results, task augmentation improves generalization performance for all the approaches. We also ablate task augmentation at different stages of training Celo, namely scheduler and parameter update rule in Table 2b. As the results suggest, task augmentation helps the most when it is added at both stages of meta-training and optimality gap (0.01) is close to the optimum (0.0). Moreover, most of the performance gain comes from adding task augmentation at the parameter update rule stage (Table 2b). This suggests that in learned optimizers, learning a robust update rule is more crucial for generalization performance than a robust scheduler.

**Learned per-parameter update rule outperforms Adam.** We experiment with replacing the learned MLP update rule in Celo with a hand-crafted rule of Adam and report its performance in Table 3a. This is equivalent to learning a scheduler over a fixed parameter update. The results clearly indicate that Celo with the learned update rule significantly outperforms the Celo with the Adam baseline, as evidenced by the IQM metric (1.20 vs. 1.06). Moreover, while Celo with Adam baseline surpasses the best-tuned Adam baseline and achieves a lower final loss (IQM > 1.0), it still lags behind the best-tuned Adam baseline on a few tasks, judging by non-zero optimality gap (0.05). This indicates that there is potential for further improving the scheduler, which could also enhance the performance of fully learned optimizers like Celo.

**Scheduler is critical for performance.** As we describe above, Celo mainly consists of two components: (1) learned update rule (2) learned scheduler. To assess the impact of scheduler on meta-generalization performance for unseen tasks, we evaluate our learned update rule from Stage 1 (§3) without the scheduler and report its performance in Table 3b. The results show that our learned update rule nearly matches the

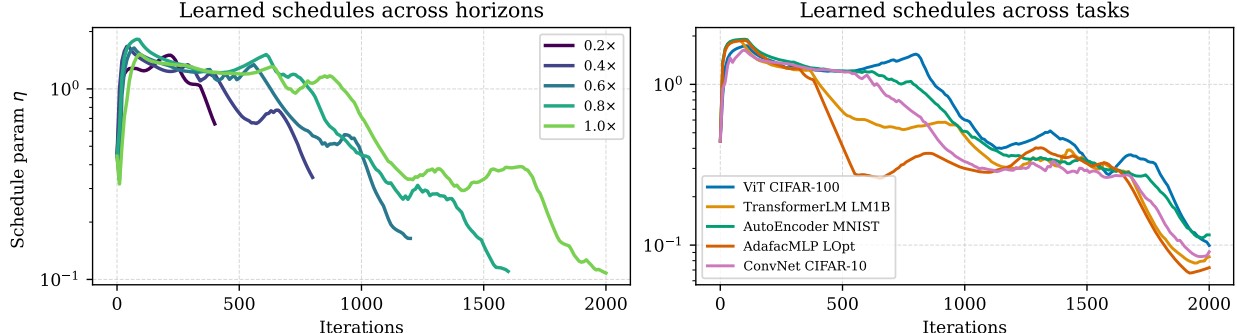

Figure 6: **Visualizing schedules predicted by Celo on unseen tasks.** Celo adapts its schedules based on the evaluation target length as shown for a ViT CIFAR-100 task (left) and also as per the given evaluation task by varying warmup and decay phases (right). Notably, Celo discovers schedules with a warmup phase followed by cyclic ramp-up and cosine decay-like phases throughout the training.

|  | IQM↑ | OG↓ |
|---|---|---|
| celo w/ adam | 1.06 | 0.05 |
| celo | **1.20** | **0.01** |

(a) **Adam vs learned update rule.** Learned parameter update rule outperforms Adam.

| celo | IQM↑ | OG↓ |
|---|---|---|
| w/o scheduler | 0.99 | 0.12 |
| w/ scheduler | **1.20** | **0.01** |

(b) **Impact of scheduler.** Using a scheduler significantly improves meta-generalization.

|  | IQM↑ | OG↓ |
|---|---|---|
| 1-stage training | 0.89 | 0.16 |
| 2-stage training | **1.20** | **0.01** |

(c) **Two-stage training.** Learning the update rule and scheduler separately in two stages is better.

| scheduler | IQM↑ | OG↓ |
|---|---|---|
| w/ tensor feats | 1.19 | 0.02 |
| w/o tensor feats | **1.20** | **0.01** |

(d) **Tensor features.** Scheduler without per-tensor features performs well.

| functional form | | IQM↑ | OG↓ |
|---|---|---|---|
| linear | $\alpha.o_t$ | 1.07 | 0.12 |
| linear + clip | $\text{clip}(\alpha.o_t)$ | 1.05 | 0.06 |
| exp | $\alpha.\exp(o_t)$ | **1.20** | **0.01** |

(e) **Functional form.** Exponential form in scheduler works the best.

Table 3: **Ablations.** Celo default settings are highlighted .

final loss of the best Adam baseline across 14 trials, achieving an IQM score close to 1.0 (0.99). However, the learned scheduler significantly improves the meta-generalization performance of Celo, resulting in an IQM score of 1.20. This improvement can be attributed to the learned scheduler's ability to respond dynamically to the unseen task (see Figure 6), enabling Celo to achieve final losses that are lower than those of Adam.

**Decoupling training of step-size scheduler and param update rule helps.** We propose to meta-train a learned optimizer in two stages, which consists of first learning a parameter update rule followed a learning a scheduler which controls the step-size in the update rule keeping the learned update rule fixed (non-trainable). We compare results of this two-stage meta-training procedure with single-stage which all the prior works follow (Metz et al., 2022b;a; 2019b) (Table 3c). Our experiments find that single-stage training quickly overfits on the meta-training tasks, specially in compute-limit setting we considered, without learning a general parameter update rule or a scheduler. Two-stage training shows better meta-generalization to unseen tasks, as indicated by the IQM in Table 3c. Two-stage training also has a superior worst-case performance (optimality gap) compared to the single-stage learned optimizer.

**Task-agnostic scheduler generalizes better.** Prior work in learned optimizers (Metz et al., 2022b; 2020b) with hierarchy uses tensor-level statistics in the scheduler to control the step size and other hyperparameters of the per-parameter update rule. However, since tensor statistics (e.g. gradient mean, momentum, etc) are highly specific for each task, learned optimizers that rely on such features often fail to generalize to newer tasks with completely different statistical properties (Almeida et al., 2021). To address this, we use

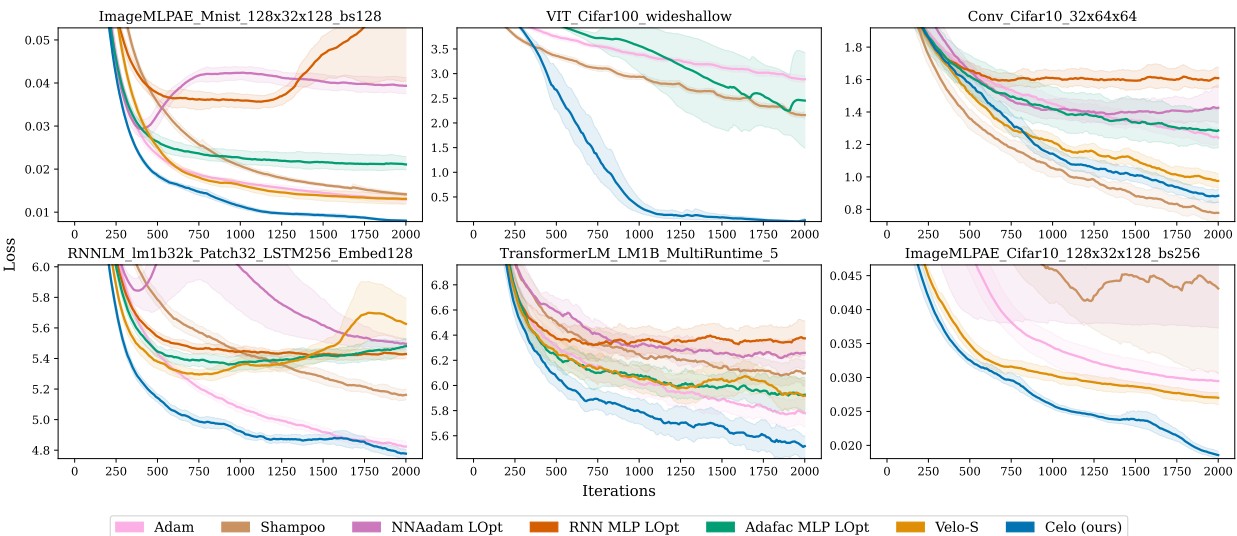

Figure 7: **Training curves for selected tasks.** See Appendix A.8 for detailed description of each task.

*task-agnostic* features such as normalized loss features and training progress which remain similar between small meta-training tasks and large-scale evaluation tasks. Table 3d compares a learned scheduler with per-tensor statistical features with Celo, which does not use any per-tensor features. The results show that our task-agnostic scheduler without any per-tensor statistics gives comparable (in fact, slightly better) results to the one which uses per-tensor statistics, at the same time being computationally cheaper. Therefore, we make it our default choice in Celo.

**Functional form of scheduler matters for generalization.** Learned scheduler plays an important during training by modulating the step-size of parameter update rule. We test different functional form for the learned scheduler in Table 3e which guides how the step-size is predicted and find that it is crucial for generalization performance. Our experiments show that exponential form performs the best and allows the scheduler to rapidly alter step-size on exponential scale which is also in line with our modern hand-designed schedulers which use cosine/exponential decays.

## 7 Related work

**Learned optimizer architectures.** Learned optimizer architectures can be broadly classified into three categories: (1) Flat learned optimizers (2) Hierarchical learned optimizers (3) Hybrid optimizers. Flat optimizers (Andrychowicz et al., 2016; Chen et al., 2020a; Metz et al., 2022a; Harrison et al., 2022) use the same update rule for every parameter in the optimizee neural network without considering any additional context such as structure of the underlying optimizee network. Hence, their parameter update computation is non-hierarchical or in other words "flat". These flat learned optimizers can be thought of as a drop-in replacement for hand-crafted update rules such as SGD or Adam. Hierarchical optimizers (Wichrowska et al., 2017; Metz et al., 2020a; 2022b; Peebles et al., 2022; Moudgil et al., 2023), unlike flat optimizers, have a hierarchical structure in their update function. They take additional meta-data of the optimizee neural network such as tensor shapes, neural graph (Kofinas et al., 2024), etc as input and utilize it in their parameter update function. These hierarchical optimizers can not only capture inter-parameter dependencies for faster optimization but also be scaled in memory-efficient manner without significantly increasing the runtime overhead since their runtime cost scale sub-linearly with the number of parameters due to hierarchy (Metz et al., 2022b;a). Hybrid optimizers (Almeida et al., 2021; Jang et al., 2023; Knyazev et al., 2024; Kristiansen et al., 2024) combine hand-crafted and learned modules in their update function. Using statistics of the underlying optimization task such as loss, gradient norms, momentum, etc, learned modules in hybrid optimizers control hyperparameters of the hand-crafted update rules such as SGD (Kristiansen et al., 2024), Adam (Metz et al., 2020b; Kristiansen et al., 2024) or LAMB (Almeida et al., 2021). Jang et al. (2023);

Knyazev et al. (2024) alternate between Adam updates and updates performed by a learned function taking only the past parameters as input showing promising results, but limited scalability. In this work, we focus on enhancing the meta-generalization capabilities of fully learned optimizers while maintaining memory and compute efficiency for practical use. Hence, our proposed approach, Celo, has a hierarchical architecture.

**Meta-generalization of learned optimizers.** Several learned optimizers have been proposed for specific settings such as physics-based motion reconstruction (Gärtner et al., 2023), reinforcement learning (Goldie et al., 2024), material design (Merchant et al., 2021), etc. Although these approaches are quite performant, their applicability to broad range of Machine Learning (ML) tasks that practitioners care about is limited. Since meta-training a learned optimizer for each setting is computationally expensive, optimizers that meta-generalize to multiple tasks after being meta-trained are more desirable. A few recent works have explored this direction. Almeida et al. (2021) proposed a hybrid optimizer with LAMB update rule that is trained on small-scale tasks and generalize to 3 orders of magnitude higher tasks. VeLO (short for "**Ve**rsatile **L**earned **O**ptimizer") is a fully learned optimizer, meta-trained using an extensive computational budget of 4000 TPU-months across diverse ML tasks. VeLO has demonstrated the ability to generalize effectively to new, unseen tasks and successfully optimize neural networks with parameters up to 600M. In this work, we propose a recipe to meta-train fully learned optimizers on a limited compute budget that meta-generalize to wide range of unseen tasks. Thérien et al. (2024) recently proposed another approach to tackle meta-generalization by developing learned optimizers within the Maximal Update Parametrization (Yang et al., 2021) framework. This enables meta-generalization to wider and deeper models as well as to longer unrolls of a meta-training task. Their approach is complementary to ours, so combining it with our recipe is a promising direction to further improve meta-generalization.

**Benchmarking optimizers.** A few works have attempted to benchmark and compare optimizers (Schneider et al., 2019; Schmidt et al., 2021; Dahl et al., 2023) on standard ML. Schneider et al. (2019) proposed the DeepOBS benchmark that contains a set of realistic 8 ML tasks and compares standard (hand-crafted) optimizers using training curves, test curves and learning rate sensitivity analysis. Schmidt et al. (2021) benchmarked 15 hand-crafted optimizers on 8 ML tasks from DeepOBS by varying tuning budget and learning rate schedules, leading to more than 50K optimization runs from all the combinations. The optimizers are compared relative to each other by using final loss/test accuracy. Dahl et al. (2023) proposed another benchmark that evaluates standard optimizers on 8 large-scale ML-Perf tasks and a methodology to score optimizers by using speedup. Speedups for each task are computed using fixed task-specific metric targets that are hand-designed. In this work, we propose evaluation metrics that can capture aggregate performance of an optimizer using both speedup and final performance criteria. Unlike prior work, our evaluation metrics are specifically designed to be robust to outliers, especially when the number of evaluation tasks is large and doesn't require hand-designing any cut-offs/thresholds. Hence, our proposed evaluation methodology is scalable with respect to number of evaluation tasks.

## 8   Limitations and future work

Our work focuses on a recipe to train learned optimizers that meta-generalize well from a limited training set. However, this work is leaving out a lot of "free" performance gains which could be obtained by trivially modifying or expanding the meta-training set, using better/faster architectures like state-space models instead of LSTMs. We intentionally kept our architecture and meta-training setup simple without any bells and whistles to do a fair head-to-head comparison with prior work and focus solely on the improvements from our recipe. Moreover, although our work significantly improves generalization performance of fully learned optimizers and we firmly believe this work is the right step towards making learned optimizers more practical, it is not focused on delivering the most-performant optimizer for all kinds of large-scale workloads. Additional work needs to be done to make these learned optimizers ready for production use cases like analyzing the impact of meta-training data distribution on generalization properties, which is out of scope for this work. Furthermore, given the amount of compute resources we had, we tried our best to include as diverse and large number of tasks as possible for evaluating all the learned optimizers and ablations with multiple seeds and hyperparameters. This resulted in hundreds of evaluation runs for each optimizer ablation. However, there is a scope for larger scale study just focused on evaluation using our proposed metrics which we leave for future work.

# 9 Conclusion

We propose a recipe to train versatile fully learned optimizers on a limited computed budget by identifying three key elements in meta-training and architecture: (1) task augmentation (2) a simple design consisting of a learned scheduler with a learned per-parameter update rule and (3) decoupled training of scheduler and per-parameter update rule. We show that our proposed learned optimizer, Celo, outperforms state-of-the-art hand-crafted and learned optimizers on unseen tasks, despite being meta-trained on a limited compute budget of 24 GPU hours, demonstrating its strong meta-generalization capabilities. Moreover, we propose evaluation metrics that can reliably capture aggregate performance of an optimizer on an evaluation set inspired from RL literature. Furthermore, we perform exhaustive ablations with meta-generalization as the principal axis of evaluation. Finally, we analyze the schedules proposed by Celo on unseen tasks and find that they are adaptive to unseen training tasks and target length horizons. Overall, we believe that this work serves as a significant stepping stone towards the development of fully learned optimizers that are efficient, hyperparameter-free, and practically applicable to large-scale unseen tasks.

## Acknowledgements

We acknowledge support from Mila-Samsung Research Grant, FRQNT New Scholar [EB], the FRQNT Doctoral (B2X) scholarship [AM] and the Canada-CIFAR AI Chair program [GL]. We also acknowledge resources provided by Compute Canada, Calcul Québec, and Mila. We thank NVIDIA for providing the GPUs that made this work possible. We are also grateful to the developers of open-source libraries such as JAX (Bradbury et al., 2018), NumPy (Harris et al., 2020), `google/learned_optimization`, and `google-research/rliable`, which were instrumental in this research.

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

# A   Appendix

## A.1   Notation table

| | |
|---|---|
| $\boldsymbol{\theta}$ | Optimizee network parameters |
| $\phi$ | Learned optimizer parameters |
| $\nabla\boldsymbol{\theta}$ | Gradient of optimizee network parameters |
| $\boldsymbol{\theta}_t$ | Optimizee network parameters at iteration $t$ |
| $\boldsymbol{s}_t$ | Optimizer state at iteration $t$ |
| $\mathcal{M}$ | Meta-data input to optimizer such as current loss, training progress, etc. |
| $\boldsymbol{\psi}_t$ | Optimizee features |
| $f_\phi$ | Function parameterized by $\phi$ |
| $\boldsymbol{F}$ | Per-param features |
| $\boldsymbol{S}$ | Normalized score matrix |
| $\mathcal{T}$ | Set of meta-training tasks |
| $\mathcal{D}$ | Inner task data |
| $\mathcal{L}$ | Inner task objective |
| $L_t$ | Loss value at iteration $t$ |
| $\eta_t$ | Scheduler scale parameter at iteration $t$ |
| $\tau$ | Task augmentation parameter |

## A.2   Additional ablations

**Results without task augmentation.**   Table 4 shows results for Celo and all the learned optimizer baselines without task augmentation as in prior work (Metz et al., 2022a; Harrison et al., 2022; Metz et al., 2020a; 2022b). As evident from the results, all the learned optimizers perform much worse for both speedup and final loss criteria when task augmentation is not used (refer to Table 1 for baseline results with task augmentation). Interestingly, VeLO-S performs better than Adafac MLP LOpt when task augmentation is not used. Moreover, VeLO-S median speedup score is 0.0 with and without task augmentation. However, it improves when task augmentation is used with respect to IQM and optimality gap metrics further highlighting the importance of our proposed metrics for tracking improvements which are not obviously captured by standard metrics like median. Except Celo, none of the learned optimizer baselines from prior work are able to achieve a final loss lower than Adam when evaluated on the 17 tasks as evident from their IQM results (final loss IQM $< 1$ and speedup $= 0$).

**VeLO ablations.**   In order to be fair in comparison with all the RNN-based learned optimizer baselines which use, we scale the hidden size of VeLO from 512 to 64 and accordingly the MLP bank size from 256 to 32 as mentioned in §5. In Table 5, we show meta-generalization results with original VeLO's original architecture (denoted by "velo") meta-trained with the same setup as other baselines (§5). Moreover, we also show results for the released pre-trained VeLO model Metz et al. (2022b) (denoted by "pretrained-velo" in Table 5) which is meta-trained using 4000 months on large collection of tasks. Note that for pre-trained VeLO, our evaluation task set is actually in-distribution (Metz et al., 2022b) but for VeLO, VeLO-S and also our proposed Celo optimizer, they are out-of-distribution. The meta-generalization performance of VeLO is slightly better with the bigger hidden size (512) than VeLO-S (IQM 0.96 vs 0.90) but still lags behind

| w/o task augmentation | final loss | | | speedup | | |
|---|---|---|---|---|---|---|
| | median↑ | OG↓ | IQM↑ | median↑ | OG↓ | IQM↑ |
| nnadam lopt (Metz et al., 2020b) | 0.62 | 0.40 | 0.65 | 0.00 | 0.90 | 0.00 |
| rnn mlp lopt (Metz et al., 2020a) | 0.77 | 0.32 | 0.77 | 0.00 | 0.88 | 0.00 |
| adafac mlp lopt (Metz et al., 2022a) | 0.93 | 0.18 | 0.89 | 0.00 | 0.78 | 0.00 |
| velo-s (Metz et al., 2022b) | 0.90 | 0.14 | 0.90 | 0.00 | 0.71 | 0.13 |
| celo (ours) | **1.14** | **0.01** | **1.20** | **1.92** | **0.14** | **1.86** |

Table 4: **Additional results without task augmentation.** All the learned optimizer baselines are meta-trained on a fixed set of 4 tasks without task augmentation (§3) as in prior work and evaluated on 17 diverse out-of-distribution tasks (refer to §5). Our proposed approach, Celo, consisting of all the three ingredients proposed in §3, significantly outperforms all the learned optimizer baselines.

our proposed approach Celo (IQM 1.20). This result shows that returns from architecture and algorithmic improvements in learned optimizers can be much higher than simply scaling optimizer sizes.

| | compute budget | IQM↑ | OG↓ |
|---|---|---|---|
| velo-s | 24 GPU hours | 0.90 | 0.14 |
| velo | 24 GPU hours | 0.96 | 0.12 |
| velo-4000 | 4000 TPU months | 1.41 | 0.00 |
| celo | 24 GPU hours | 1.20 | 0.01 |

Table 5: **VeLO ablations.** IQM and optimality gap (OG) metrics for final loss criteria on our 17 evaluation task set (§5) are reported along with meta-training compute budget. Note that our evaluation task set is *in-distribution* for pretrained-velo but out-of-distribution for celo and all the other velo baselines.

**Task augmentation level ablations.** Task augmentation i.e. re-parametrization can be applied at different hierarchical levels of neural network during meta-training: (1) Global level, where a single augmentation parameter is sampled for the entire network; (2) Tensor level, where an augmentation parameter is sampled for each tensor or layer in the network; (3) Per-parameter level, where an augmentation parameter is sampled individually for each parameter in the network; and (4) Mixed strategy, where the augmentation parameter is sampled uniformly from a combination of the previous three approaches. In our experiments, for mixed augmentation strategy, we first randomly sample a strategy from the list ["none", "none", "global", "global", "tensor", "tensor", "parameter"] and then randomly select an augmentation range from the list [[0.001, 1000.0], [0.01, 100.0], [0.1, 10.0]]. Additionally, different augmentation strategies can used in the two stages for our approach presented in §3. We present results for all the ablations in Table 6. Overall, we find that global augmentation performs the best closely followed by tensor augmentation.

| | task aug level | | | |
|---|---|---|---|---|
| | stage 1 | stage 2 | IQM↑ | OG↓ |
| | tensor | global | 1.13 | 0.04 |
| | global | tensor | 1.13 | 0.04 |
| | mix | mix | 1.14 | 0.03 |
| | tensor | tensor | 1.18 | 0.04 |
| celo | global | global | **1.20** | **0.01** |

Table 6: **Task augmentation level ablations.** IQM and optimality gap (OG) metrics for final loss criteria on our 17 evaluation task set (§5) are reported. Task augmentation is applied at different levels during two-stage meta-training proposed in §3. "mix" refers to randomly sampling from a mixture of level list, see § A.2 for more details.

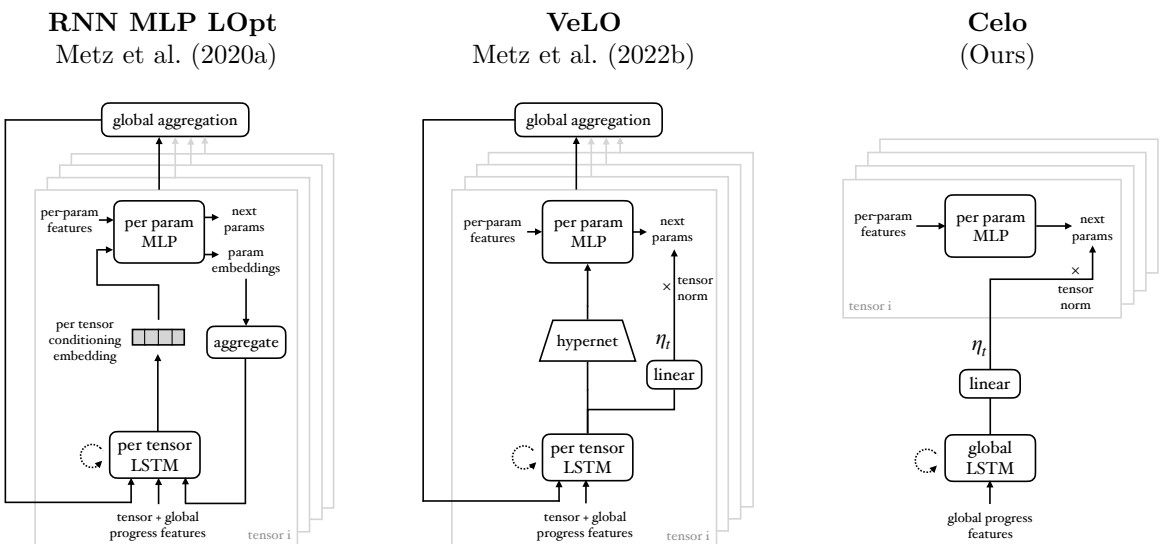

Figure 8: **Comparing Celo architecture with VeLO and RNN MLP LOpt.** Our proposed approach, Celo, not only simplifies learned optimizer architecture from prior work as shown above but also vastly outperforms in meta-generalization performance (see §1).

## A.3 Implementation details

We meta-train and evaluate all the learned optimizers in this work in JAX (Bradbury et al., 2018) using the open-sourced learned optimization[2] library. For fair comparison, all the learned optimizers are meta-trained with exactly same PES setup: maximum inner unroll length 2000 with unroll lengths sampled logarithmically sampled between 100 and 2000, standard deviation 0.01, truncation length 50 and mean inner training loss as the meta-objective. We use Adam (Kingma & Ba, 2015) as the meta optimizer and meta-train for 100K iterations on a single Nvidia RTX8000 GPU. Following prior work (Metz et al., 2022a; Harrison et al., 2022), we meta-train each learned optimizer using AdamW as the meta-optimizer by sweeping over 3 seeds and 5 learning rates [3e-5, 5e-5, 1e-4, 3e-4, 1e-3]. We find that higher and lower learning rates often result in unstable training with PES, hence learning rates such as 1e-4 and 3e-4 work the best for all learned optimizers (except NNAdamW LOpt for which lower learning rates, 3e-5 and 5e-5, work better). Meta-training jobs for all the learned optimizers in this work finish within 24 hours. In order to keep learned optimizers hyperparameter-free and do fair comparison, we do not use weight decay in learned optimizers although it has been shown to help for some tasks (Metz et al., 2022b; Harrison et al., 2022) but requires additional tuning. We evaluate all the learned optimizers in this work on our 17 task evaluation set with 3 seeds per task. For task augmentation in meta-training, we sample a random parameter logarithmically between 0.001 and 1000 in each meta-iteration and use it to re-parametrize optimizee network weights at the global level in inner-training (Metz et al., 2022b). As noted in prior work (Metz et al., 2022b), this augmentation can simulate even more tasks when it is applied at different levels in the optimizee neural network: global level, tensor level and per-parameter level by sampling global (one $\tau$ for all parameters), per-tensor or per-parameter augmentations ($\tau$s) respectively. In our experiments, we find out that global-level augmentation performs the best with respect to meta-generalization and hence we use it as the default. Moreover, in order to decrease the impact of noise and loss fluctuations during training (e.g. lucky batches with low loss values or loss spike at the end of training), we smooth the loss curves for Adam baseline and optimizers being evaluated using exponential moving average with coefficient 0.9 before computing the normalized scores. We compute final loss by taking average over multiple batches (10) as in prior work (Metz et al., 2022b;a). We use the official code `rliable`[3] open-sourced by (Agarwal et al., 2021) for computing IQM and other metrics.

---

[2]https://github.com/google/learned_optimization/

[3]https://github.com/google-research/rliable/

### A.4 Scaling study

In our main paper, learned optimizers are meta-trained with a 2K unroll length. To study scaling behavior of our proposed approach, we meta-train Celo and VeLO-S with unroll lengths of 1K, 2K, and 5K, then meta-test them on all tasks in our evaluation set using a 10K unroll length, assessing generalization to longer unrolls. We report the IQM metric with the final loss criterion and show scaling trends in the right Figure. Celo performance scales predictably with the meta-training budget and gets IQM 1.85 on 10K unroll length with 5K meta-train unroll length, hence demonstrating strong generalization performance with respect to unroll length. VeLO-S, on the other hand, gets IQM 0.34 with 5K meta-train length. These results clearly demonstrate that Celo scales significantly better than VeLO, achieving much better performance under a fixed meta-training budget, thereby highlighting its compute efficiency.

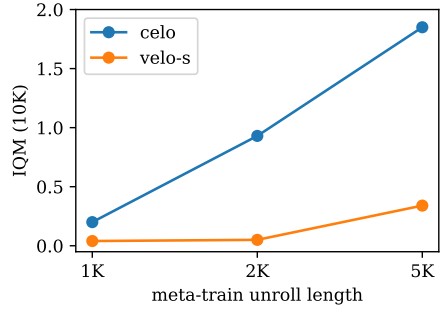

### A.5 Runtime analysis

Celo updates the parameters of a given optimizee network using a per-parameter MLP. Since the number of FLOPs for a forward pass of a linear layer with weights $n \times m$ is $2nm$, the cost of a 2-layer MLP update rule in Celo with 30 input features, 2 hidden layers with 4 units and 2 outputs units is $2 \times (30 \times 4 + 4 \times 4 + 4 \times 2) = 288$ FLOPs per parameter. For an optimizee linear layer with $n \times m$ parameters and batch size $B$, the total amount of FLOPs to perform a forward and backward pass is $6nmB$. Hence, the ratio of FLOPs of our learned optimizer's update w.r.t. to forward+backward FLOPs is $288nm/6nmB = 48/B$. This shows that the overhead of our optimizer decreases as the tasks get larger with more inputs flowing through the optimizee network with larger batch size $B$ which also makes sense intuitively because as the tasks become computationally expensive, the cost of forward and backward pass dominates optimizer update cost. In this analysis, we omitted the cost of LSTM scheduler but as noted by prior work (Metz et al., 2022b;a), the overhead of any tensor or global-level computation in the learned optimizer decreases as the number of parameters increase in the optimizee network since the cost of per-parameter updates dominate and tensor/global computation overhead asymptotically reduces to a fixed constant (Metz et al., 2022b). We compare wall-clock time of Celo with Adam and other learned optimizer baselines in Table 7. We benchmark runtime on a NVIDIA V100 GPU with 10 seeds per optimizer.

### A.6 MLCommons AlgoPerf evaluation

We additionally evaluate on a few standard ML Commons tasks from AlgoPerf benchmark (Dahl et al., 2023) which are much larger and completely out-of-distribution for meta-trained optimizers in this work. Figures 9 and 10 show training and validation curves respectively, comparing Celo-Adam (Celo scheduler with Adam update rule discussed in Table 3) with Schedule-Free AdamW, the winning entry in the self-tuning track of AlgoPerf 2024 competition as well as tuned NAdamW baseline provided by the AlgoPerf team (Kasimbeg et al., 2025). For baselines, we directly use the results data from the AlgoPerf 2024 competition. AlgoPerf is a time-to-result benchmark, where runs are halted either upon reaching the task's validation target or upon timing out. As evident from the results, Celo-Adam manages to optimize these relatively large-scale unseen workloads despite being meta-trained on small image MLP tasks (§A.7). However, the final validation performance of Celo-Adam still lags behind the tuned baselines and it fails to hit validation targets earlier as evident from the validation curves (Figure 10). Morever, we find that Celo, pre-trained on simple image MLP tasks considered in this work, is unstable for MLCommons tasks, hence we omit it from Figures 9 and 10. This highlights that learning a robust update rule is critical to scale fully learned optimizers to large-scale workloads since learned schedulers can generalize, as evident from Celo-Adam results, even from limited meta-training. We defer the scaling up of meta-training for Celo to future work, which we believe is the right direction to make it ready for standard use-cases especially given that our proposed recipe is compute-efficient and scalable (A.4).

| evaluation tasks | runtime per step (ms) | | | |
|---|---|---|---|---|
| | adam | velo-s | velo | celo |
| ImageMLP_FashionMnist_Relu128x128 | 0.20 | 1.69 | 1.76 | 1.31 |
| ImageMLP_Cifar10_128x128x128_LayerNorm_Relu | 0.31 | 3.41 | 3.83 | 2.69 |
| ImageMLP_Cifar10_128x128x128_Tanh_bs128 | 0.24 | 2.74 | 3.05 | 2.20 |
| Conv_Cifar10_32x64x64 | 1.12 | 2.80 | 3.02 | 2.44 |
| Conv_Cifar10_32x64x64_batchnorm | 1.29 | 3.68 | 4.10 | 3.22 |
| Conv_Cifar10_32x64x64_layernorm | 1.41 | 3.78 | 4.11 | 3.21 |
| Conv_Cifar100_32x64x64 | 1.11 | 2.70 | 3.09 | 2.55 |
| VIT_Cifar100_wideshallow | 6.73 | 43.89 | 46.00 | 39.47 |
| VIT_Cifar100_skinnydeep | 4.52 | 29.09 | 32.69 | 23.70 |
| ImageMLPAE_Cifar10_128x32x128_bs256 | 0.44 | 4.29 | 4.28 | 4.16 |
| ImageMLPAE_Mnist_128x32x128_bs128 | 0.19 | 2.54 | 2.70 | 2.22 |
| RNNLM_lm1b32k_Patch32_LSTM256_Embed128 | 23.96 | 53.42 | 53.38 | 52.86 |
| RNNLM_wikipediaen32k_Patch32_LSTM256_Embed128 | 24.29 | 53.47 | 53.81 | 53.38 |
| TransformerLM_LM1B_MultiRuntime_0 | 0.71 | 8.30 | 8.71 | 7.10 |
| TransformerLM_LM1B_MultiRuntime_2 | 0.99 | 14.50 | 15.46 | 12.94 |
| TransformerLM_LM1B_MultiRuntime_5 | 1.26 | 10.93 | 11.42 | 9.88 |
| LOpt_AdafacMLPLOpt_FashionMnist_50 | 65.12 | 67.31 | 68.16 | 67.06 |

Table 7: **Runtime per step for each task (ms).** Celo is faster in terms of wallclock time per step than previous state-of-the-art learned optimizers such as VeLO (Metz et al., 2022b) as well as VeLO with smaller hidden size (VeLO-S from §5). However, compared to Adam, Celo has additional overhead depending on the evaluation task. Notably, as tasks become more computationally expensive, this overhead diminishes, as discussed in §A.5.

## A.7    Meta-training tasks

We meta-train all the learned optimizers in this work on a set of small image MLP tasks from prior work (Metz et al., 2022b) in order to keep the meta-training fast and perform all the ablation experiments within our compute budget. The same task set is also used by VeLO to compare different learned optimizer architectures (Metz et al., 2022b). It consists of the 4 tasks with different datasets namely, Fashion MNIST, CIFAR-10, MNIST and SVHN. All the tasks share the same configuration which consists of a 1-layer MLP network with 32 hidden units, $8 \times 8$ single channel image input, batch size 64 and ReLU activations. Link to code snippets here.

## A.8    Evaluation tasks

We evaluate Celo and all the optimizer baselines on the following 17 tasks. Each task name contains link to its associated code snippet.

1. ImageMLP_FashionMnist_Relu128x128: Image classification task with Fashion MNIST dataset, MLP network with 2 hidden layers each containing 128 hidden units and ReLU activations, batch size 128, cross-entropy loss.

2. ImageMLP_Cifar10_128x128x128_LayerNorm_Relu: Image classification task with CIFAR-10 dataset, MLP network with 3 hidden layers each containing 128 hidden units with layer-norm followed by ReLU in each hidden layer, batch size 128, cross-entropy loss.

3. ImageMLP_Cifar10_128x128x128_Tanh_bs128: Image classification task with CIFAR-10 dataset, MLP network with 3 hidden layers each containing 128 hidden units with Tanh activations, batch size 128, cross-entropy loss.

4. Conv_Cifar10_32x64x64: Image classification task with CIFAR-10 dataset, convolutional neural network with 3 hidden layers containing 32, 64 and 64 channels with ReLU activations, stride 2 in first layer and stride 1 in rest of the layers, batch size 128, cross-entropy loss.

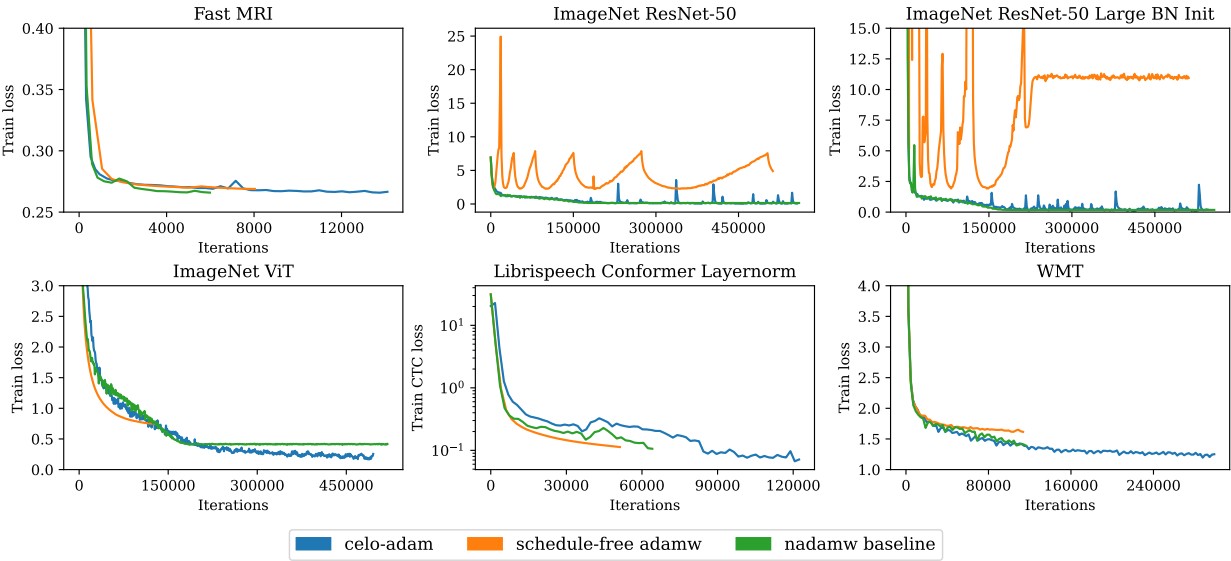

Figure 9: **MLCommons AlgoPerf train curves.**

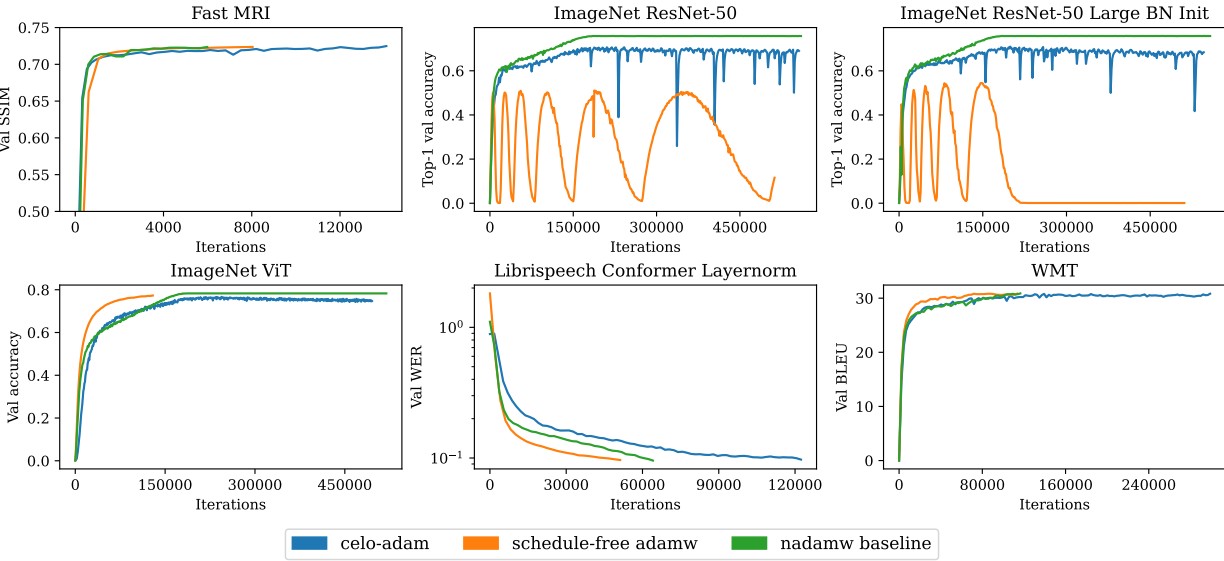

Figure 10: **MLCommons AlgoPerf validation curves.** We compare Celo-Adam with winner in the self-tuning track of AlgoPerf 2024 competition Schedule-Free AdamW and prize qualification baseline NAdamW which is heavily tuned (Kasimbeg et al., 2025; Dahl et al., 2023). Although Celo-Adam is just meta-trained on small image MLP tasks (§A.7), it manages to optimize much larger tasks, in some cases, even better than Schedule-Free AdamW (ImageNet ResNet tasks). However, it still lags behind baselines in terms of final performance, a gap that can be potentially covered by improving and scaling up meta-training.

5. `Conv_Cifar10_32x64x64_batchnorm`: Image classification task with CIFAR-10 dataset, convolutional neural network with 3 hidden layers containing 32, 64 and 64 channels with ReLU activations and batch-norm, stride 2 in first layer and stride 1 in rest of the layers, batch size 128, cross-entropy loss.

6. `Conv_Cifar10_32x64x64_layernorm`: Image classification task with CIFAR-10 dataset, convolutional neural network with 3 hidden layers containing 32, 64 and 64 channels with ReLU activations and layer-norm, stride 2 in first layer and stride 1 in rest of the layers, batch size 128, cross-entropy loss.

7. `Conv_Cifar100_32x64x64`: Image classification task with CIFAR-100 dataset, convolutional neural network with 3 hidden layers containing 32, 64 and 64 channels with ReLU activations and layernorm, stride 2 in first layer and stride 1 in rest of the layers, batch size 128, cross-entropy loss.

8. `VIT_Cifar100_wideshallow`: Image classification task with CIFAR-100 dataset, vision transformer model with 6 transformer layers each containing 6 attention heads, 16×16 patch input, hidden size 384, MLP expand factor 4, batch size 128, cross-entropy loss.

9. `VIT_Cifar100_skinnydeep`: Image classification task with CIFAR-100 dataset, vision transformer model with 10 transformer layers each containing 4 attention heads, 16×16 patch input, hidden size 128, MLP expand factor 4, batch size 128, cross-entropy loss.

10. `TransformerLM_LM1B_MultiRuntime_0`: Language modeling task with LM1B dataset, transformer decoder model with 1 layer containing 5 attention heads, hidden size 20, sequence length 8, vocabulary size 32K, batch size 4, cross-entropy loss.

11. `TransformerLM_LM1B_MultiRuntime_2`: Language modeling task with LM1B dataset, transformer decoder model with 2 layers each containing 4 attention heads, hidden size 32, sequence length 8, vocabulary size 32K, batch size 8, cross-entropy loss.

12. `TransformerLM_LM1B_MultiRuntime_5`: Language modeling task with LM1B dataset, transformer decoder model with 1 layers each containing 4 attention heads, hidden size 32, sequence length 32, vocabulary size 32K, batch size 8, cross-entropy loss.

13. `ImageMLPAE_Cifar10_128x32x128_bs256`: Auto-encoder task with CIFAR-10 dataset, MLP network with shape 128-32-128 and ReLU activations, batch size 256, MSE loss.

14. `ImageMLPAE_Mnist_128x32x128_bs128`: Auto-encoder task with MNIST dataset, MLP network with shape 128-32-128 and ReLU activations, batch size 128, MSE loss.

15. `RNNLM_lm1b32k_Patch32_LSTM256_Embed128`: Language modeling task with LM1B dataset, LSTM recurrent network with embedding size 128, hidden size 256, sequence length 32, vocabulary size 32K, batch size 128, cross-entropy loss.

16. `RNNLM_wikipediaen32k_Patch32_LSTM256_Embed128`: Language modeling task with wikipedia dataset, LSTM recurrent network with embedding size 128, hidden size 256, sequence length 32, vocabulary size 32K, batch size 128, cross-entropy loss.

17. `LOpt_AdafacMLPLOpt_FashionMnist_50`: Learned optimizer meta-training task using Fashion MNIST dataset, MLP-based learned optimizer (Adafac MLP LOpt), 1-layer optimizee network with 32 hidden units, $8 \times 8$ image input, maximum inner unroll length 50, outer batch size 2, inner loss cross-entropy, mean loss of inner loop as the meta-objective.

## A.9 Evaluation plots

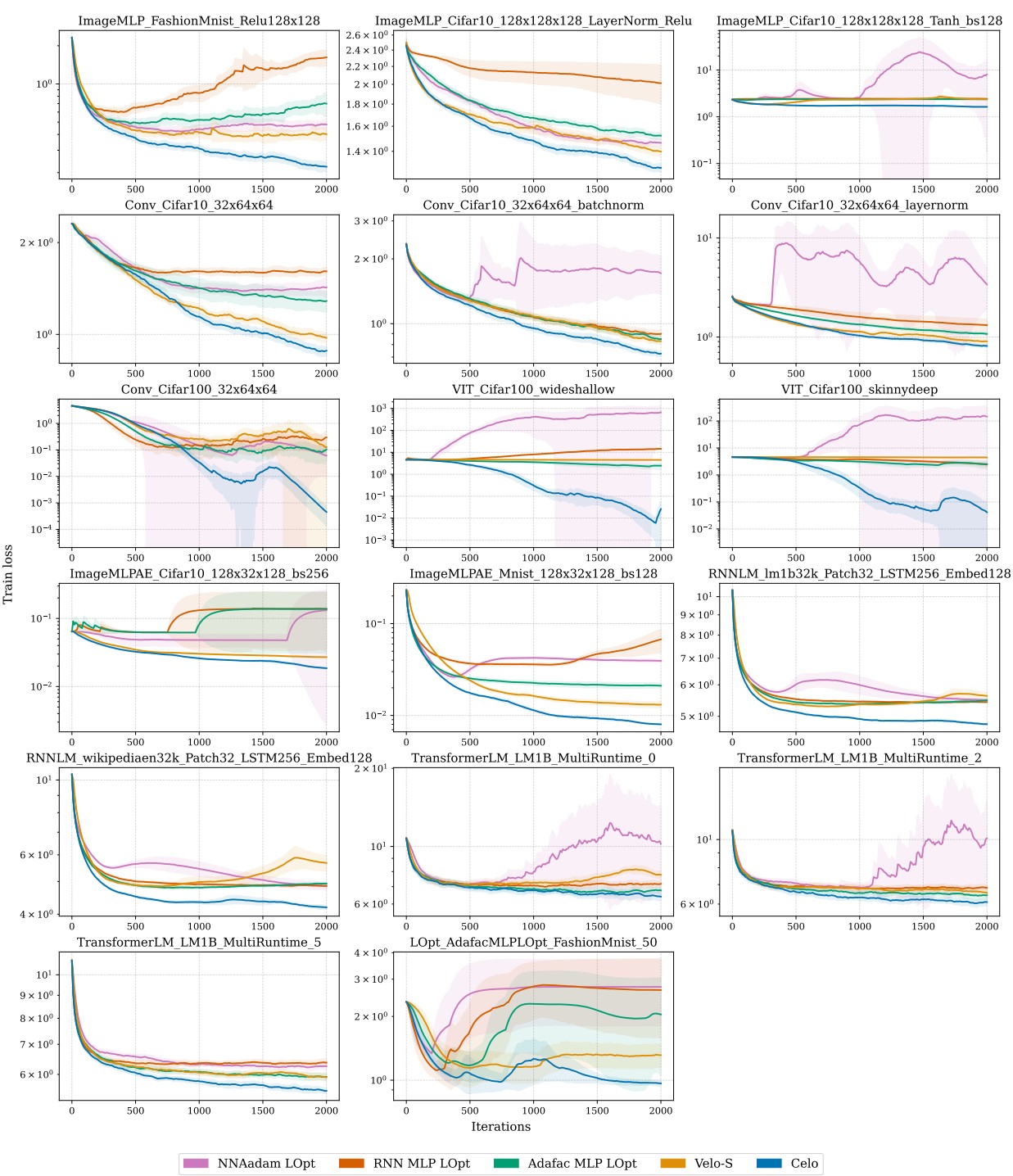

Figure 11: **All train curves.**

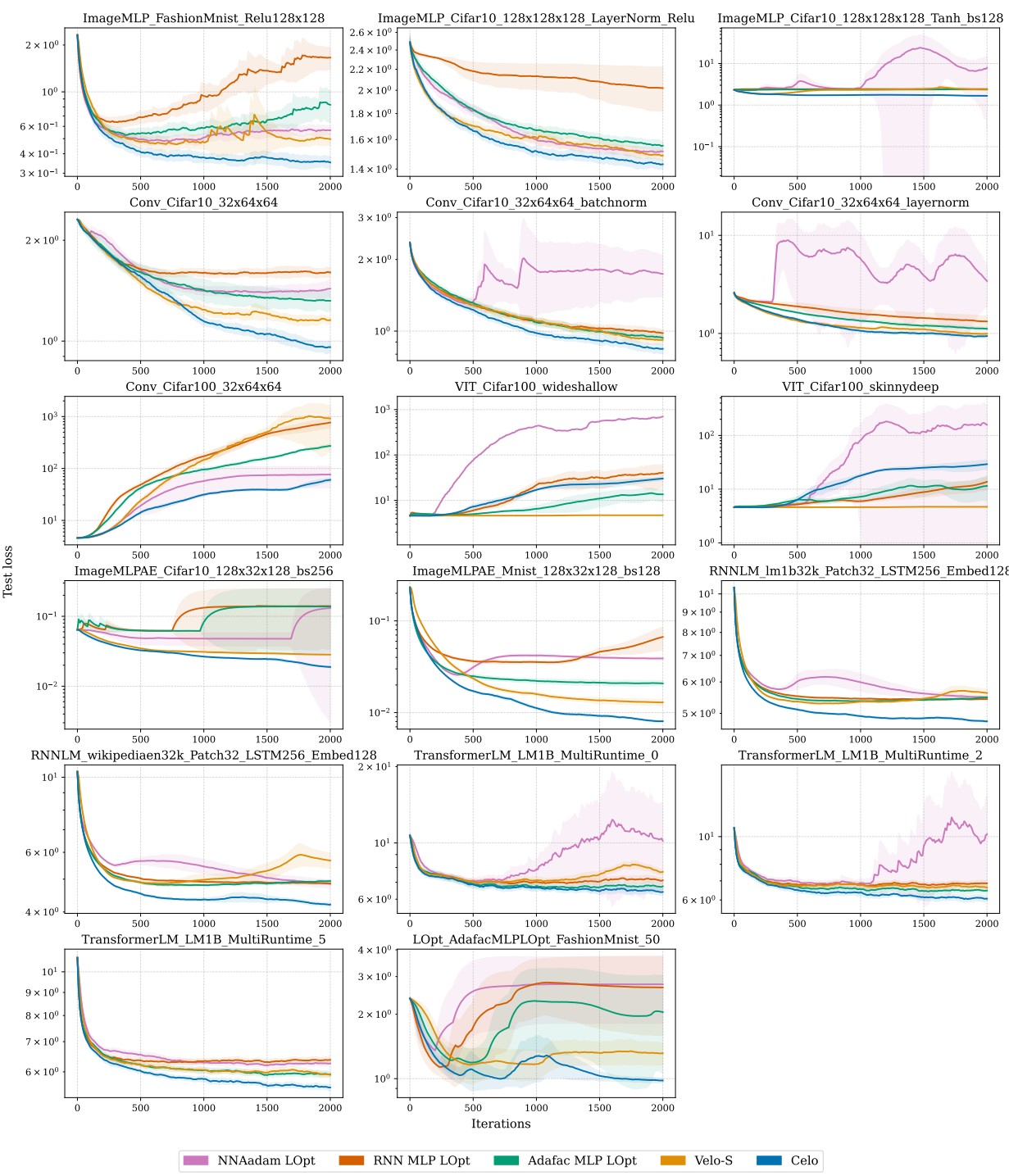

Figure 12: **All test curves.**

## A.10    Evaluation plots with task augmentation

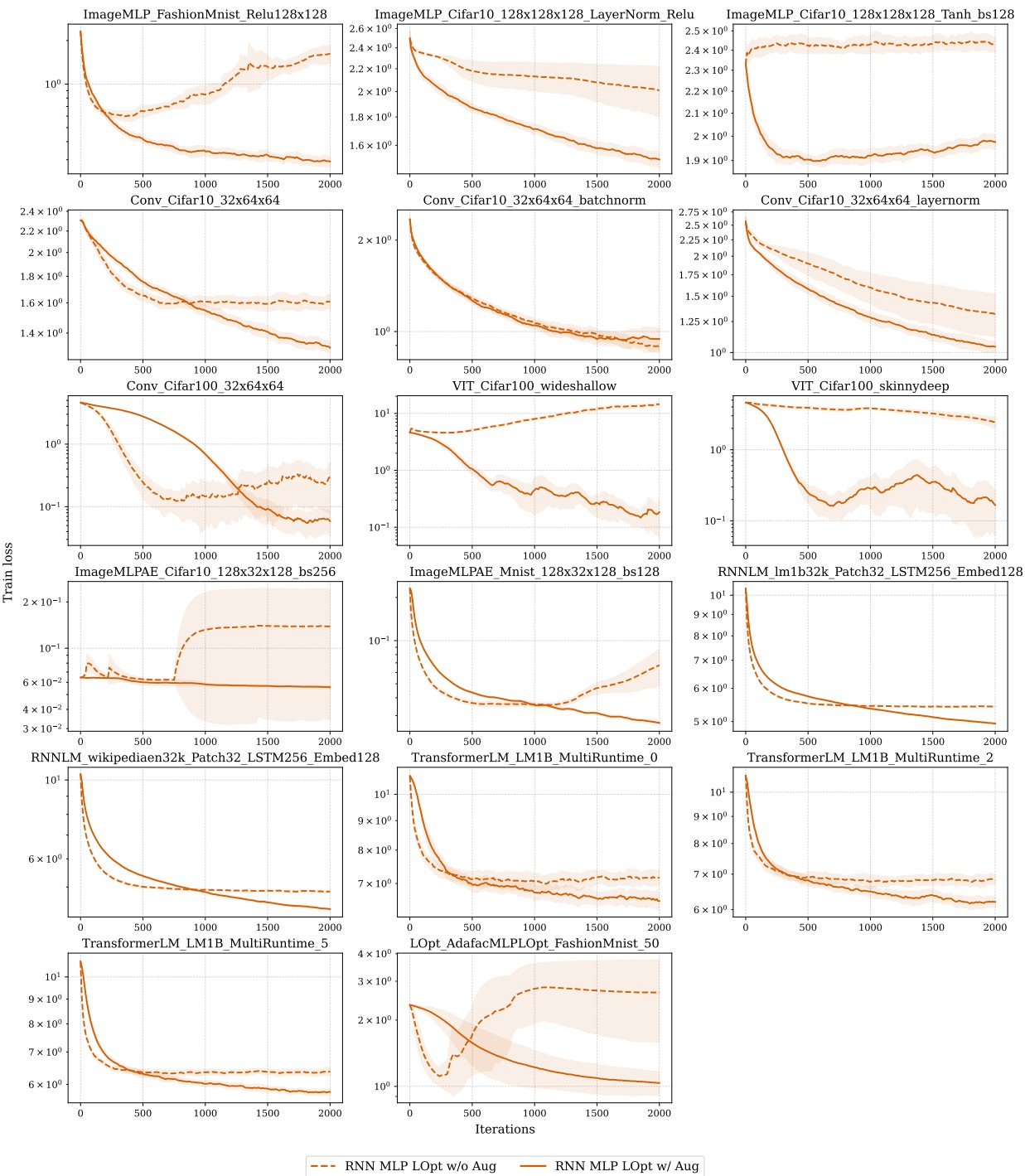

Figure 13: **RNN MLP LOpt augmentation training curves.**

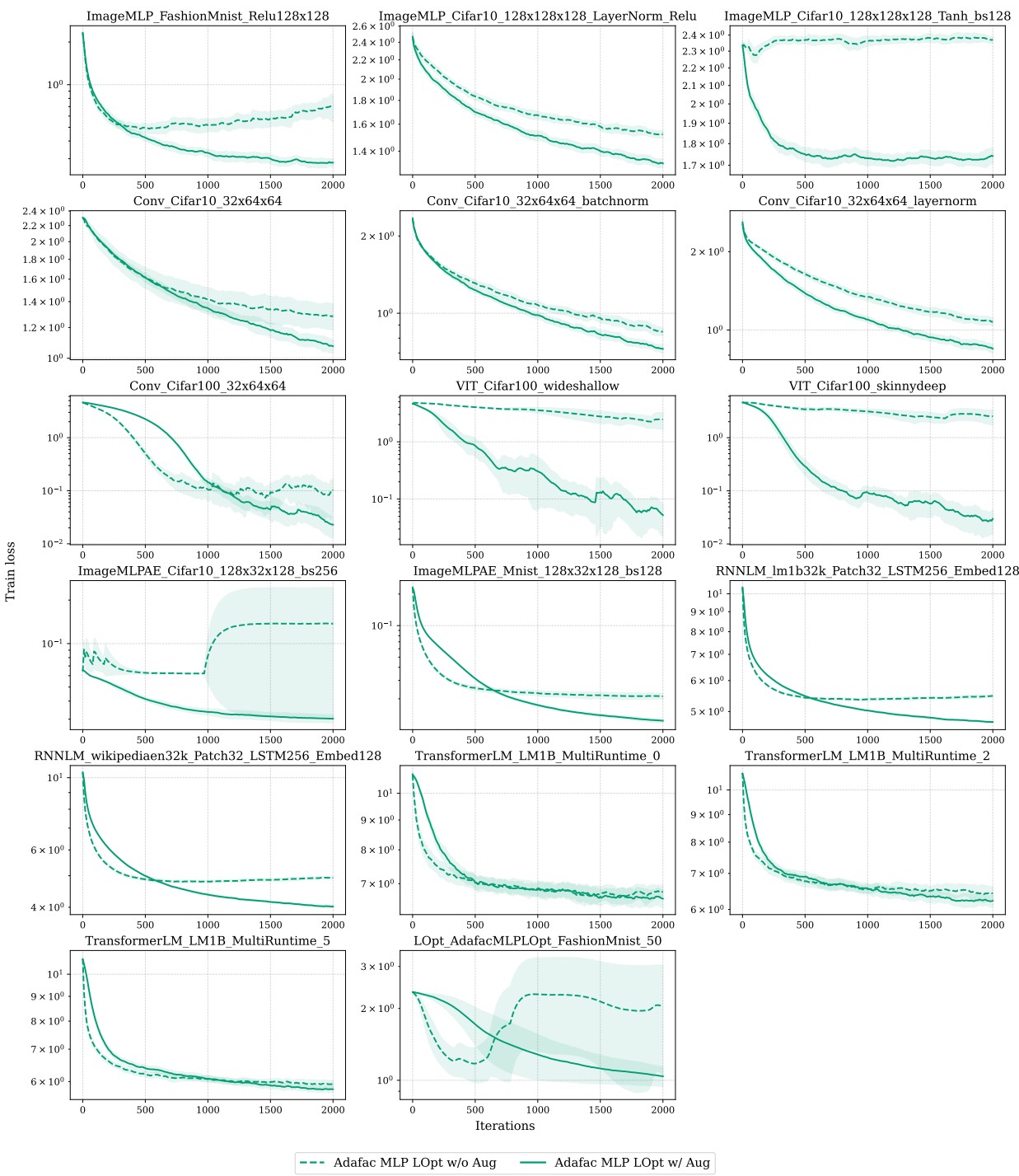

Figure 14: **Adafac MLP LOpt augmentation training curves.**

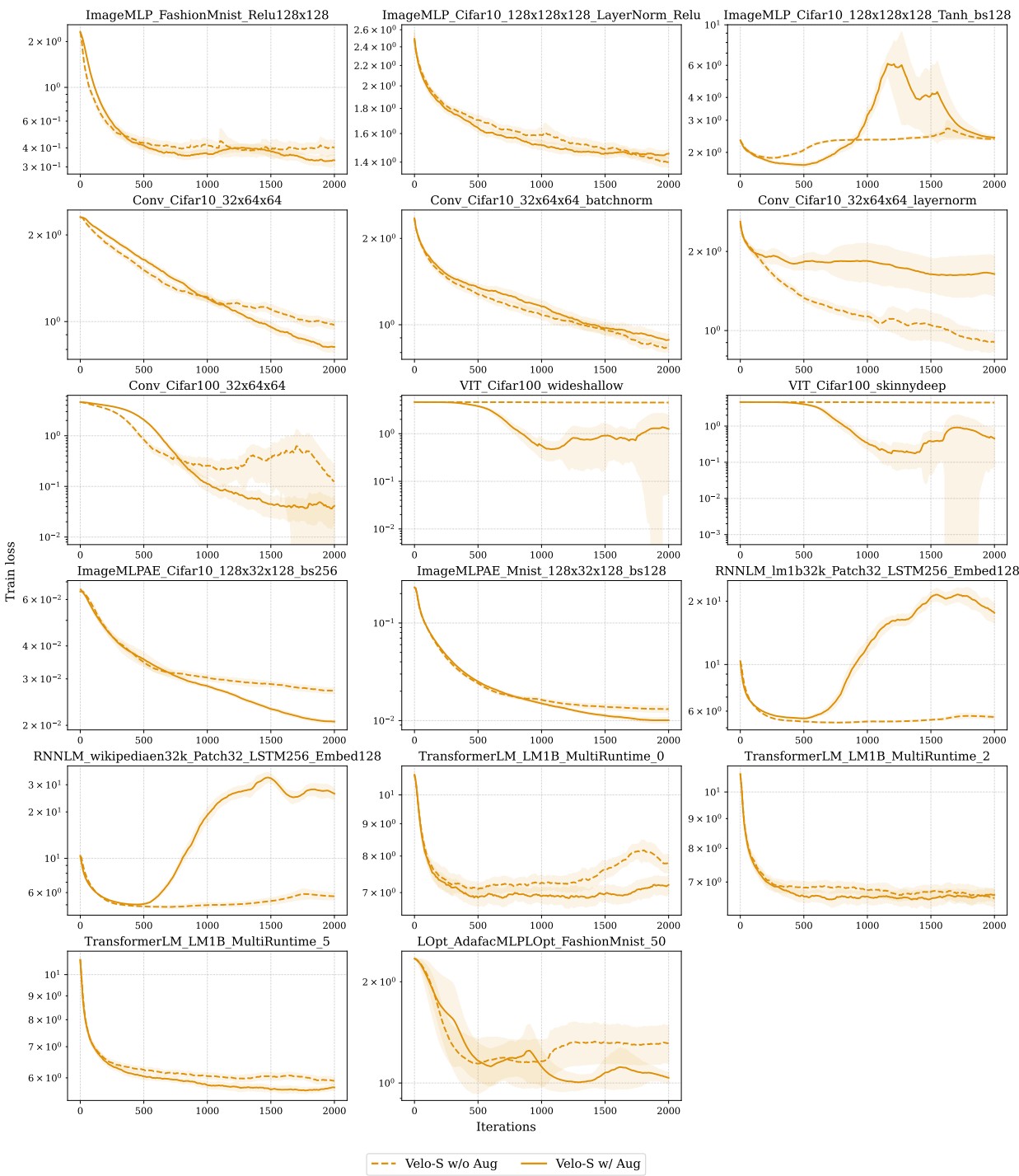

Figure 15: **Velo-S augmentation training curves.**

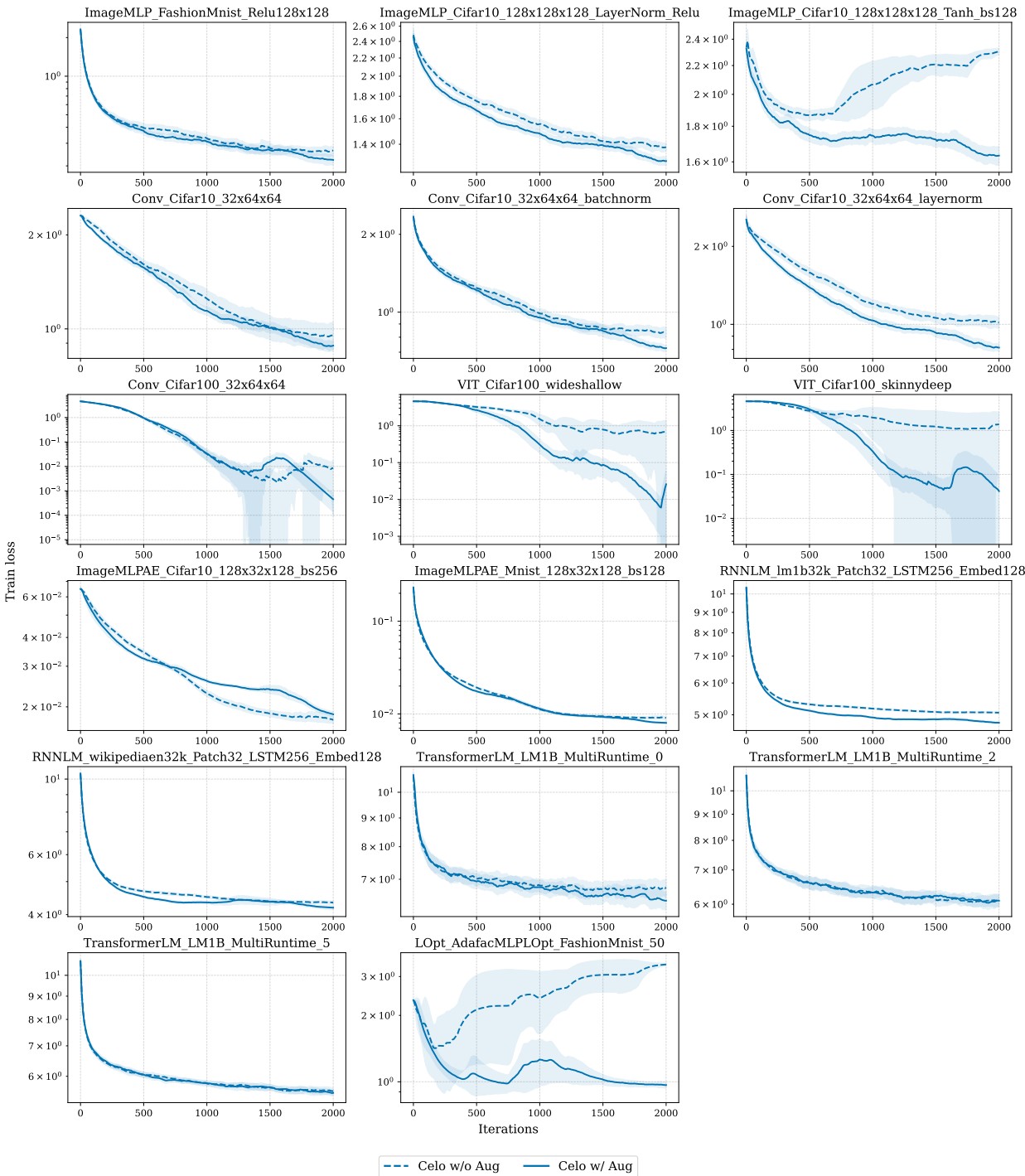

Figure 16: **Celo augmentation training curves.**

## A.11 Celo schedules

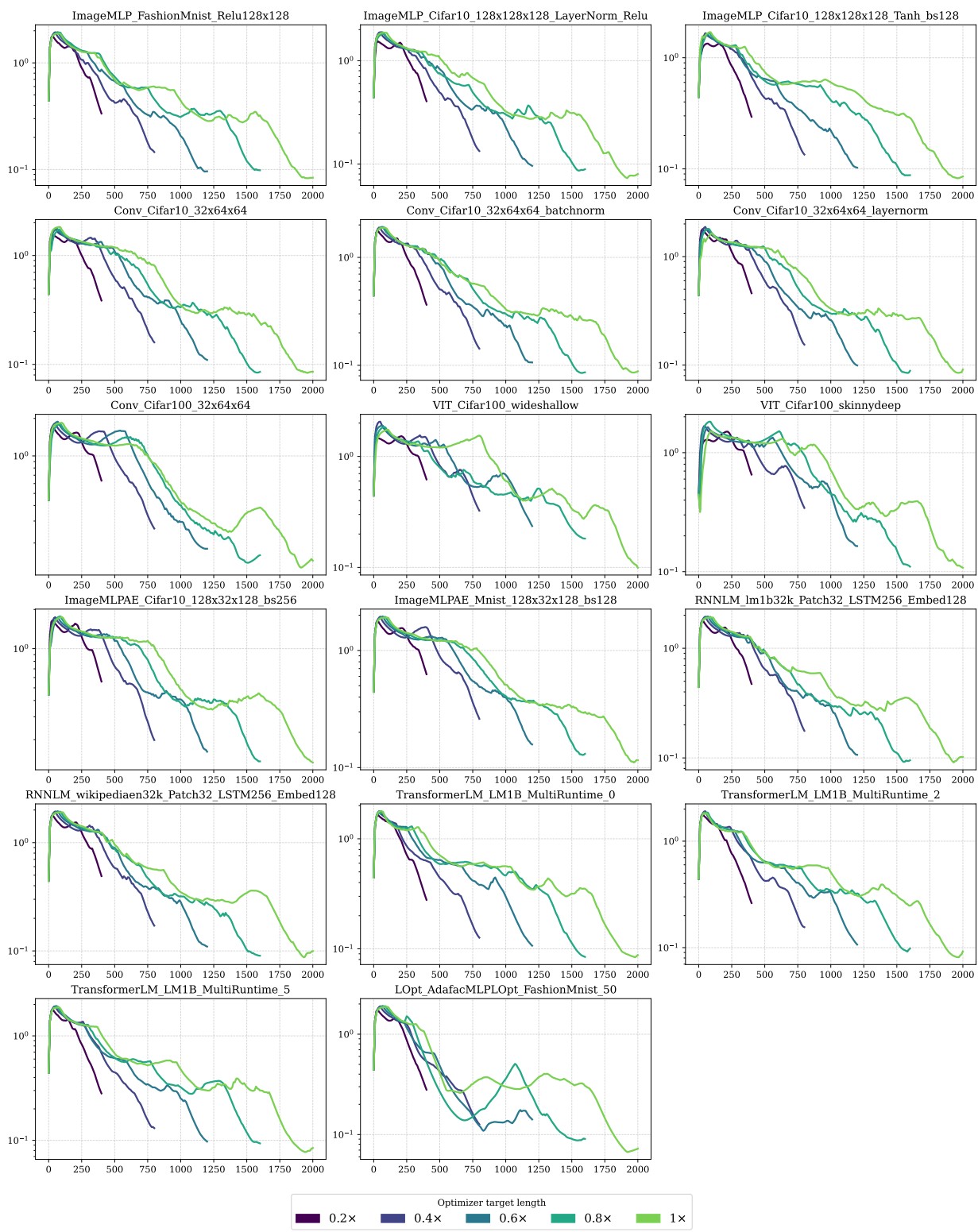

Figure 17: **Celo learned schedules for all tasks.**

