# OpenReview forum: "Celo: Training Versatile Learned Optimizers on a Compute Diet"
_TMLR — Accepted by TMLR_

### Review · Reviewer_Yef6 · 2025-02-11

**Summary Of Contributions:**

* Data: Data augmentation to effectively simulate new tasks by doing a reparameterization trick on the optimizee's parameters.

* Optimization: Decoupled, multi-stage training of per-parameter (local) updates and scheduler scalar (global) learning rates.

* Metrics: Final & speedup IQM scores for the task: Inspired from reinforcement learning (Agarwal et al., 2021), the authors replace the human baseline with an Adam baseline, normalize the metric of interest by the baseline, and sum over the meta-learning tasks and the meta-learning steps.

**Audience:**

Yes

**Claims And Evidence:**

Yes

**Requested Changes:**

We need to see costs of the optimization runs in Figure 4 (and corresponding result artifacts).

minor:

linear projecting <- linearly projecting

best Adam baseline <- the best Adam baseline

lowest loss <- the lowest loss

Etc. (please check typos and grammar)

**Strengths And Weaknesses:**

Strengths:

* 4000h TPU -> 24h GPU is an impressive achievement, but I think the results need more comparison about the actual cost of the competitor optimizers.

* The discovery that "task augmentation" is key for generalization.

* Outperforming a strong suite of learned and handcrafted optimizers.

* Nice set of ablations in Table 3.

Weaknesses:

* The experiments are not scaled, so the gains from the current scales may not transfer to larger scales.

---

> ### Author Response · Authors · 2025-03-27
> **Response to Reviewer Yef6**
>
> We thank the reviewer for their constructive feedback, we address the main concerns below:
>
> **The experiments are not scaled, gains may not transfer.** We understand the reviewer's concern. We'd like to point out that _our recipe is general and scalable_, even though the pre-trained optimizers in this work may not directly generalize to all kinds of large-scale workloads due to limited meta-training. To validate this, we conducted a scaling study with respect to meta-training budget (Section A.4 in Appendix) and show that as the meta-training budget is increased, Celo's meta-generalization performance improves consistently and significantly outperforms the VeLO baseline. To be specific, we meta-train Celo and VeLO-S with unroll lengths of 1K, 2K, and 5K, then meta-test them on all tasks in our evaluation set using a 10K unroll length, assessing generalization to longer unrolls. We score the optimizers using the final loss IQM metric. Celo achieves final loss IQM 1.85 with 5K length compared to VeLO-S which gets IQM 0.34, hence demonstrating its strong generalization performance (5x boost). We leave a more extensive study on scaling for future work.
>
> **Costs of the optimization runs in Figure 4.** We have added another section A.5 in Appendix on runtime analysis containing theoretical estimation of FLOP overhead of Celo along with a table with observed runtime per step for all the evaluation tasks. As we also pointed to Reviewer j6jn -- even though learned optimizers perform more computation than standard hand-designed optimizers such as SGD and Adam, their overhead generally decreases as tasks become more computationally expensive and batch size increases, since the cost of forward and backward pass dominate the parameter update cost of optimizers. For a standard ResNet-50 task with batch size 32, the overhead multiplier of learned MLP update rule wrt SGD is 1.32x [1] which should further decrease with larger batch size and for large-scale PaLM model with 540B params, overhead of VeLO (which is more expensive than Celo) reduces to roughly 2% [2]. However, for a non-standard task, the overhead of learned optimizer may depend on various factors such as layer type, batch size, depth, width in the optimizee neural network and even hardware/kernel implementation; the trends sometimes aren't that obvious (see Figure 4 in [1]).
>
> **Typo and grammar fixes.** We thank the reviewer for taking a detailed look, we have thoroughly reviewed and fixed all the issues in the revised draft.
>
> [1] Metz, Luke, et al. "Practical tradeoffs between memory, compute, and performance in learned optimizers." Conference on Lifelong Learning Agents (CoLLAs). PMLR, 2022.
> [2] Metz, Luke, et al. "VeLO: Training versatile learned optimizers by scaling up, 2022." URL https://arxiv.org/abs/2211.09760.

---

### Review · Reviewer_j6jn · 2025-02-20

**Summary Of Contributions:**

The paper proposed a new meta-optimizer that can be meta-trained on a small computation budget. They proposed a decoupled design of optimizer and scheduler, and a unique way for performing task augmentation for better meta-generalization. The proposed algorithm is able to meta-optimize from a small number of datasets and on a toy model architecture using a very small number of GPU hours. The resulting meta-optimizer outperforms existing optimizers on many unseen datasets and models.

**Audience:**

Yes

**Claims And Evidence:**

Yes

**Requested Changes:**

- Add wallclock time
- Add more standard benchmarks (e.g. ImageNet) with standard architecture (e.g. ResNet 50)

**Strengths And Weaknesses:**

Strengths:
- The major selling point of the paper is that it is able to meta-generalize from a few datasets and toy models to unseen datasets and large models.
- The experiments are thorough. The paper ablated settings such as learned update, scheduler, two stage training, etc. The paper evaluates MLP, conv net, and LM transformers on different datasets and benchmarks.
- Task augmentation is a novel idea, and is empirically effective.
- The experiments contain many unseen test datasets ran with multiple random seeds.

Weaknesses and Questions:
- The metrics are based on the number of iterations, but the learned optimizer could incur more computation. It would be good to clarify on the wall clock time implication.
- It is surprising that the paper is able to generalize so well on 8x8 images with a one-layer MLP network. It could also be interesting to test what is the limit on such meta-training and meta-generalization. Can any datasets work? Can the datasets be synthetic?
- Lastly, I would love to see if there are some standard benchmarks with a standard architecture, where we can reach a known accuracy (e.g. ImageNet) with less number of iterations. The CNN models evaluated in the paper are fairly small.

---

> ### Author Response · Authors · 2025-03-27
> **Response to Reviewer j6jn**
>
> We thank the reviewer for their constructive feedback, we address the main concerns below:
>
> **Wall clock time.** We have included an additional section A.5 on runtime analysis in Appendix containing theoretical estimation of FLOP overhead of Celo along with a table with observed runtime per step for all the evaluation tasks. Even though learned optimizers perform more computation than standard hand-designed optimizers such as SGD and Adam, their overhead generally decreases as tasks become more computationally expensive and batch size increases, since the cost of forward and backward pass dominate the parameter update cost of optimizers. For a standard ResNet-50 task with batch size 32, the overhead multiplier of a learned MLP update rule (similar to one we use in Celo) wrt SGD is 1.32x [1] which should further decrease with larger batch size and for large-scale PaLM model with 540B params, overhead of VeLO (which is more expensive than Celo) reduces to roughly 2% [2]. However, for a non-standard task, the overhead of learned optimizer may depend on various factors such as layer type, batch size, depth, width in the optimizee neural network and even hardware/kernel implementation; the trends sometimes aren't that obvious (see Figure 4 in [1]).
>
> **Standard task such as ResNet50 Imagenet.** We thank the reviewer for the suggestion. We have included 6 tasks from MLCommons AlgoPerf benchmark [3] including two ResNet-50 ImageNet tasks which are much larger and completely out-of-distribution for learned optimizers considered in this work which are meta-trained on small image MLP tasks and much shorter unroll horizon (Section A.6). The training and validation curves are presented in Figures 9 and 10 in Appendix which compare with Schedule-Free AdamW, the winner from the self-tuning track in AlgoPerf 2024 competition as well as NAdamW baseline tuned exhaustively by the AlgoPerf team. Our variant of Celo, namely, Celo-Adam (learned scheduler with Adam update rule, also discussed in Table 5), manages to optimize relatively stably and even better in ResNet tasks than Schedule-Free AdamW. However, the final validation performance of Celo-Adam still lags behind Schedule-Free AdamW and fails to hit validation targets earlier as evident from the validation curves (Figure 10). We find that Celo and learned optimizer baselines to be unstable on these AlgoPerf tasks due to their limited meta-training budget, hence we omit them from the comparison. These AlgoPerf results show that learned schedulers can meta-generalize even from a very limited meta-training set when paired with a robust update rule such as Adam. They also suggest that the primary bottleneck for the meta-generalization capability of fully learned optimizers lies in the learned update rule and making it more robust should significantly improve meta-generalization performance of learned optimizers. Our preliminary scaling study in Appendix A.4 shows that our recipe scales significantly better than previous methods such as VeLO and achieves much better generalization performance for a given budget. As indicated in the Scope and Limitations section, we leave the scaling up of meta-training to future work to make Celo ready for standard use cases.
>
> **Synthetic meta-training dataset.** We tried meta-training on a set of 4 noisy quadratic multi-dimensional problems with dimensions 2, 10, 100 and 1000 using our proposed recipe. Note that a quadratic task with 1000 dimensions is in the same order of magnitude in terms of param count as our meta-training image MLP task which contains roughly 3K params and they only differ in optimization dynamics. We find that Celo meta-trained on this set achieves final IQM 0.02 and Optimality Gap (OG) 0.89, which is much worse compared to meta-training on image MLP tasks (IQM 1.20, OG 0.01). These results indicate that meta-training tasks should be somewhat reflective of meta-test distribution. A targeted study on the effect of meta-training task distributions on generalization performance is feasible, which we leave for future work.
>
> [1] Metz, Luke, et al. "Practical tradeoffs between memory, compute, and performance in learned optimizers." Conference on Lifelong Learning Agents (CoLLAs). PMLR, 2022.
> [2] Metz, Luke, et al. "VeLO: Training versatile learned optimizers by scaling up, 2022." URL https://arxiv.org/abs/2211.09760.
> [3] Dahl, George E., et al. "Benchmarking neural network training algorithms." arXiv preprint arXiv:2306.07179 (2023).

---

### Review · Reviewer_2iuG · 2025-03-13

**Summary Of Contributions:**

The paper introduces Celo, a compute-efficient learned optimizer designed to achieve strong meta-generalization using minimal computational resources. By identifying key ingredients such as task augmentation, a decoupled two-stage training procedure (first learning a parameter update rule, then a scheduler), and simplified architecture, the proposed optimizer significantly outperforms prior learned and hand-designed optimizers on diverse out-of-distribution tasks. Additionally, the authors introduce robust evaluation metrics derived from reinforcement learning literature to reliably measure performance and generalization of optimizers.

**Audience:**

Yes

**Claims And Evidence:**

Yes

**Requested Changes:**

Please address W1 and W2.

Besides, authors are recommended to check more benchmark settings in "Learning to Optimize: A Primer and A Benchmark" (JMLR 2022)

**Strengths And Weaknesses:**

S1 - Clear Methodological Innovation: The paper proposes an elegant and efficient L2O architecture named "Celo," carefully designed to be computationally efficient while achieving strong generalization. The methodology clearly identifies and effectively addresses critical issues like truncation bias by employing a two-stage training strategy, clearly separating update rule learning from scheduler learning. This allows the optimizer to achieve state-of-the-art performance on diverse out-of-distribution tasks, demonstrating robustness and efficiency in practical scenarios.

S2- The authors provide extensive experimental results, including comparisons with both learned and hand-designed optimizers across various optimization scenarios. The introduction of robust evaluation metrics derived from reinforcement learning literature, such as meta-generalization measures, provides a more reliable assessment of optimizer quality beyond traditional benchmarks. These careful, thorough experiments add substantial credibility to the claims.

S3 - The methodological innovations are supported by theoretical justifications, particularly regarding the convergence properties and generalization behaviors of the learned optimizer. By clearly identifying critical aspects such as truncation bias, and proposing targeted strategies like curriculum-based training, the paper significantly advances theoretical understanding in the Learning to Optimize (L2O) community.

W1 -  The proposed two-stage training and task augmentation methods bear close resemblance to established curriculum and imitation learning approaches introduced in "Training Stronger Baselines for Learning to Optimize" (Chen et al., NeurIPS 2020). Despite this similarity, the submission did not discuss, cite, or directly compare against this prior work.

W2 - While task augmentation is central to achieving strong generalization, the submission does not rigorously investigate how variations in the task distributions (e.g., complexity or diversity of augmented tasks) directly influence performance. A targeted ablation study explicitly varying augmentation settings, along with analysis of sensitivity to task sampling strategies, would be essential to validate claims and strengthen methodological rigor.

---

> ### Author Response · Authors · 2025-03-27
> **Response to Reviewer 2iuG**
>
> We thank the reviewer for their constructive feedback, we address the main concerns below:
>
> **Discussion about Training Stronger Baselines for Learning to Optimize (NeurIPS 2020).** We thank the reviewer for pointing us to this work, we have added it to the related work section. We highlight below the differences compared to Chen et al. Our work builds on the line of work by Metz et al. [1, 2, 3, 4] that uses Persistent Evolutionary Strategies; it already gives unbiased gradient estimate and does not suffer from truncation bias, a key problem that is handled empirically by Chen et al. through curriculum learning. Moreover, Chen et al. focuses on learning by imitating hand-designed optimizers which improves generalization capabilities of learned optimizers as evident from their experiments, but we believe the potential of their approach is still limited by hand-designed optimizers. Experiments in their work focus mainly on improving performance over previous learned optimizer approaches, and they compare with tuned standard optimizers such as Adam, only for MNIST tasks. Our work, on the other hand, allows learning optimizers that can not only surpass hand-designed optimizers through evolution-based learning but also generalize to unseen tasks and outperform tuned hand-designed optimizers. However, techniques presented in their work such as curriculum and imitation-based learning can still be beneficial in our framework, hence we have cited it in the related work section. Finally, we emphasize that our work is contextualized and compared with the most recent state-of-the-art (SOTA) method, VeLO [5]. The settings and applications presented in another paper pointed by the reviewer "Learning to Optimize: A Primer and A Benchmark (JMLR 2022)" will be interesting to consider in the future as we mostly focus on deep learning tasks in our work.
>
> **Task augmentation ablations.** We have included additional experiments related to different sampling strategies as suggested by the reviewer in Appendix Table 6 (Section A.2). Specifically, we apply task augmentation at different levels in the optimizee neural network during meta-training such as global, tensor, per-parameter and mixture of these. We find that the default global augmentation which we use in the main paper performs the best. As mentioned in the Scope and Limitations section, we leave investigating the effect of meta-training task distributions on generalization performance to future work. In this work, our goal is to push the meta-generalization capabilities of learned optimizers given a fixed meta-training dataset. We have included an additional scaling study in Appendix Section A.4 to show that our recipe scales better and predictably than previous approaches as we increase the meta-training budget.
>
> [1] Metz, Luke, et al. "Understanding and correcting pathologies in the training of learned optimizers." International Conference on Machine Learning. PMLR, 2019.
> [2] Vicol, Paul, Luke Metz, and Jascha Sohl-Dickstein. "Unbiased gradient estimation in unrolled computation graphs with persistent evolution strategies." International Conference on Machine Learning. PMLR, 2021.
> [3] Metz, Luke, et al. "Practical tradeoffs between memory, compute, and performance in learned optimizers." Conference on Lifelong Learning Agents. PMLR, 2022.
> [4] Harrison, James, Luke Metz, and Jascha Sohl-Dickstein. "A closer look at learned optimization: Stability, robustness, and inductive biases." Advances in neural information processing systems 35 (2022): 3758-3773.
> [5] Metz, Luke, et al. "VeLO: Training versatile learned optimizers by scaling up, 2022." URL https://arxiv.org/abs/2211.09760.

---

### Decision · Action_Editor_3hkq · 2025-05-01

**Recommendation:** Accept as is

**Comment:**

This paper introduces a new meta-optimizer called Celo that appears to train much faster than competing alternatives due to various ideas explored in the paper including task augmentation, and use of learnt schedulers and update rules. It outperforms 15 hand-designed optimizers, including Adam as well as other meta-learning baselines in comprehensive experiments.

**Audience:**

Yes, since efficient optimization interests a broad ML/AI audience

**Claims And Evidence:**

The experiments are comprehenshive, with ablations such as learned update, scheduler, two stage training, etc. The paper evaluates MLP, conv net, and LM transformers on different datasets and benchmarks. Overall, the proposed method is convincingly benchmarked.